# Choice history effects in mice and humans improve reward harvesting efficiency

**Junior Samuel López-Yépez** [1,2¤], **Juliane Martin** [2], **Oliver Hulme** [3,4], **Duda Kvitsiani** [2]*

**1** Aarhus University, Department of Chemistry, Aarhus, Denmark, **2** Aarhus University, Danish Research Institute of Translational Neuroscience (DANDRITE), Aarhus, Denmark, **3** Danish Research Centre for Magnetic Resonance, Centre for Functional and Diagnostic Imaging and Research, Copenhagen University Hospital Hvidovre and Amager, Hvidovre, Denmark, **4** London Mathematical Laboratory, London, United Kingdom

¤ Current address: CortAIx,Thales, Montreal, Canada
* kvitsi@dandrite.au.dk

**Data Availability Statement:** Location of the data is at url www.figshare.com The url of each dataset is provided here: Mouse data for Fig 2 https://figshare.com/articles/dataset/Mice_1-7/14540283 Human data for Fig 2 https://figshare.com/articles/

## Abstract

Choice history effects describe how future choices depend on the history of past choices. In experimental tasks this is typically framed as a bias because it often diminishes the experienced reward rates. However, in natural habitats, choices made in the past constrain choices that can be made in the future. For foraging animals, the probability of earning a reward in a given patch depends on the degree to which the animals have exploited the patch in the past. One problem with many experimental tasks that show choice history effects is that such tasks artificially decouple choice history from its consequences on reward availability over time. To circumvent this, we use a variable interval (VI) reward schedule that reinstates a more natural contingency between past choices and future reward availability. By examining the behavior of optimal agents in the VI task we discover that choice history effects observed in animals serve to maximize reward harvesting efficiency. We further distil the function of choice history effects by manipulating first- and second-order statistics of the environment. We find that choice history effects primarily reflect the growth rate of the reward probability of the unchosen option, whereas reward history effects primarily reflect environmental volatility. Based on observed choice history effects in animals, we develop a reinforcement learning model that explicitly incorporates choice history over multiple time scales into the decision process, and we assess its predictive adequacy in accounting for the associated behavior. We show that this new variant, known as the double trace model, has a higher performance in predicting choice data, and shows near optimal reward harvesting efficiency in simulated environments. These results suggests that choice history effects may be adaptive for natural contingencies between consumption and reward availability. This concept lends credence to a normative account of choice history effects that extends beyond its description as a bias.

dataset/Human_Data_VI_task/16676740 Mouse
data for Fig 3: Volatility Fig 3A and 3B https://
figshare.com/articles/dataset/Volatility/14540346
Difference in set reward probabilities Fig 3C and 3D
https://figshare.com/articles/dataset/Difference_in_
set_reward_probabilities/14540298, Difference in
sum reward probabilities Fig 3E and 3F https://
figshare.com/articles/dataset/difference_in_sum_
of_set_reward_probabilities/14540349.

**Funding:** D.K. and J.M received funding under
grant number DANDRITE-R248-2016-2518 from
Lundbeck Foundation, https://lundbeckfonden.
com. J.S.L. received funding under grant number
R191-2015-1506 from Lundbeck Foundation,
https://lundbeckfonden.com. The funders had no
role in study design, data collection, analysis,
modelling, or decision to submit the work for the
publication.

**Competing interests:** The authors have declared
that no competing interests exist.

## Author summary

Animals foraging for food in natural habitats compete to obtain better quality food
patches. To achieve this goal, animals can rely on memory and choose the same patches
that have provided higher quality of food in the past. However, in natural habitats simply
identifying better food patches may not be sufficient to successfully compete with their
conspecifics, as food resources can grow over time. Therefore, it makes sense to visit from
time to time those patches that were associated with lower food quality in the past. This
demands optimal foraging animals to keep in memory not only which food patches pro-
vided the best food quality, but also which food patches they visited recently. To see if ani-
mals track their history of visits and use it to maximize the food harvesting efficiency, we
subjected them to experimental conditions that mimicked natural foraging behavior. In
our behavioral tasks, we replaced food foraging behavior with a two choice task that pro-
vided rewards to mice and humans. By developing a new computational model and sub-
jecting animals to various behavioral manipulations, we demonstrate that keeping a
memory of past visits helps the animals to optimize the efficiency with which they can har-
vest rewards.

## Introduction

Numerous perceptual and decision-making tasks have shown that choices systematically
depend on the history of past choices, often referred to as choice history bias [1–8]. This phe-
nomenon is often framed as a bias rather than an adaptive phenomenon because it typically
degrades the rate of reward consumption obtained by animals in experimental contexts. How-
ever, these studies predominantly make the implicit assumption that environmental resources
are decoupled from the choices that animals make. In other words, such studies assume that
past choices have no bearing on a future reward availability. Such an assumption is often vio-
lated by virtue of the causal structure of natural habitats. For instance, foraging for food in one
patch may deplete the patch's energy resources in the short term, whereas a patch left unvisited
for longer period of time may allow for an increased likelihood of replenishment. Thus, past
choices impact the local foraging environment and affect outcome statistics. The variable
interval (VI) reward schedule task is a simple task that circumvents this problem by making
reward availability contingent on choice history [9]. In each trial of the discrete version of the
VI task, equally sized rewards are assigned with fixed or set reward probabilities attached to
different options. Such options could constitute the left or right port entry for mice, or the left
or right arrow key press on a keyboard for humans. Once assigned, the reward remains avail-
able until the subject chooses the associated option and obtains the reward. After a reward is
obtained, it is not available until it is assigned to the same option. As a result, each option's
reward probability is defined by both its initial henceforth, its set reward probability (probabil-
ity unique for each option), and the number of trials without that option being chosen. Thus,
reward probabilities can change as a function of the animal's past choices. In other words, the
probability of earning a reward, which is conditional on choosing a given option, is a function
of the number of trials that have elapsed since that option was last chosen. Choosing the option
with the highest set reward probability does not maximize the reward rate in this task, because
even if the option has a lower set reward probability, the longer the time since that option was
last chosen the higher the chance that the reward will be assigned to a given option. Therefore,
after a certain period, the probability that a reward is assigned to an unchosen option escalates
until the option becomes that with the highest reward probability.

The best strategy for maximizing the reward rate over all of the available options thus depends on not only the set reward probability of each option, but also the history of how recently each option has been chosen in the past. To maximize their reward rate, optimal agents should choose the options with the highest set reward probability and switch when the reward probability of the other unchosen options overtakes that of the initial option. This choice strategy requires optimal agents to guide their decisions based on both past choices and set reward probabilities. However, whether animals use a choice strategy consistent with optimal models of behavior is an open question.

Evaluating whether animals follow optimal foraging principles faces several challenges. This is because, in experimental settings that approximate natural foraging, many variables such as energy, time, and opportunity costs are difficult to measure, and competing models can generate qualitatively similar predictions [10–12]. These problems can be mitigated in reward foraging tasks that require subjects to initiate decisions from identical starting points to equally distant options. This ensures that all else being equal, different choices bear the same costs for the decision maker. Because of this, we were permitted to formulate optimal models of behavior.

Here, we used a VI task to study foraging behavior from a normative perspective. We acquired behavioral data from both humans and mice and compared the choice history effects that we observed to those generated by optimal agents. By manipulating first- and second-order statistics of the reward outcomes, we characterized the contingency between choice and reward availability that may drive choice history effects. This result prompted us to derive a local decision rule by incorporating the choice history effects into a reinforcement learning (RL) algorithm. We call this the double trace (DT) model because it models choice history with both fast and slow decaying exponential functions. We found that choice history effects observed in species as diverse as mice and humans and including published data from monkeys [13], improved reward harvesting efficiency in the VI task. Thus, we provide an explanatory account of choice history effects beyond their description as a bias. This finding potentially connects choice history phenomena to a broader class of optimality models within behavioral ecology.

## Materials and methods

### Ethics statement

All the experiments on mice were approved by the Danish Animal Experiments Inspectorate under the Ministry of Justice (Permit 2017—15—0201—01357) and were conducted according to institutional and national guidelines. Human behavioral task was not considered as health science research project and did not require approval by the Science Ethics Committee of Central Jutland Region, Denmark.

### Rodent behavioural tasks

Water-deprived male mice (C57Bl/6J strain) were trained to initiate the trials by poking their noses into a central port equipped with sensors (infrared light emitter and phototransistor) that detected the exact time of entry and exit of animals. After poking their noses into the central port, the mice were free to poke their noses into the left or right sides of the port. Rewards were delivered inside each port via a metal tube using solenoid valves (www.theleeco.com) according to the VI schedule following Eq (1). There was no fixed inter-trial interval. Mice could initiate the next trial immediately after poking their noses into one of the side port. We did not use any punishment or time-out period. Before testing mice on the VI task, mice underwent two training stages. In the first stage mice could obtain water rewards (hence

rewards) by poking any of the three ports. This stage consisted of 20 trials. In the second stage, mice were required to poke their nose into the port that was lit. Light was delivered by light emitting diodes, mounted inside each port. The correct choice of the port resulted in water delivery. Once animals learned to correctly identify the lit port in $> 80\%$ of trials the animals were classified as ready to run the VI task. The training of animals took 2–7 days. The mice were tested inside a dark chamber in their day phase of the circadian cycle. The behavioral chamber and nose ports were 3D printed (www.shapewayes.com). The behavioural protocols were written in MATLAB (www.mathworks.com) which controlled an Arduino-based system Bpod (www.sanworks.io). 267 sessions using eleven mice (18–47 sessions per mouse) were recorded in total. On average each mouse completed 610 trials per session (standard deviation (SD) of 359 trials). Each session lasted on average 39 minutes (SD of 18 minutes). The duration of each session was determined by the performance of each mouse. If mice did not initiate a new trial within 10–15 minutes, the session was terminated.

## Human behavioural tasks

The VI task was implemented in the form of a computer game, and performed by 19 individuals (aged 18–60, 12 identified as females, 6 as males and 1 as other). The computer game was written on the Unity platform (www.unity.com), in which an installation package was distributed via email to participants. The performance of each participant was collected by email. The virtual task environment consisted of two apple trees and a gate separating the participant's avatar from the trees. The player controlling the avatar had to first open the gate (delay intervals: 0–1s, 2–3s and 4–5s) before subsequently choosing one of the trees. On each trial apple tree provided one apple with the probability that followed the VI task schedule (Eq (1)). Prior to playing the game each participant received brief instructions (see below) on how to play the game. They were, however, not informed about the specifics of the reward schedules or the task structure. Each participant completed at least six blocks consisting of 20–30 trials (mean trials per session = 239, SD = 135). Within each block, the task conditions (gate intervals and reward probability) remained constant but changed randomly between blocks.

## Optimal agents

While optimal agents do not necessarily offer biologically realistic models of choice, they are still informative both conceptually, and in terms of providing upper bounds of performance given their specific constraints and assumptions. We constructed the following three optimal agents:

**The Oracle agent.** The choices of an Oracle agent follow a simple rule: Choose the option $i$ with the highest probability of reward on each trial ($a = argmax_{i \in A} P(R|i, t)$). The update of this probability follows the rule:

$$P(R|i, t + 1) = 1 - (1 - P_{set}(R|i, t))(1 - P(R|i, t)(1 - \delta_{i,t})). \tag{1}$$

$\delta_{i,t} = 1$ when an agent chooses option $i$ at time $t$; $\delta_{i,t} = 0$ otherwise.

**The inference-based (IB) agent.** The IB agent was constructed to infer the set reward probabilities $P_{set}(R|i, t)$ of each side $i \in \{1, 2\}$ at trial $t = 1, 2, \ldots, T$ while taking actions and observing rewards $R \in \{1, 0\}$ in a VI task. We assume that the agent "knows" the structure of the VI task Eq (1) and follows that equation to update reward probabilities. We ran this agent by initializing with $N = 1000$ random estimates of the set reward probabilities $\hat{\theta}_n(i)$, where $n = 1, 2, \ldots, N$, drawn from a uniform probability distribution $Pprior(\hat{\theta}(i))$ with boundaries at

0 and 1. On every trial $t$, it estimated the reward probabilities,

$$\hat{P}_n(R) = 1 - (1 - \hat{\theta}_n(i))^{T_i},\tag{2}$$

where $T_i$ is the number of trials since the option $i$ was chosen. Then, it generated the value of each option as

$$value_i = \frac{1}{N}\sum_{n=1}^{N}\hat{P}_n(R).\tag{3}$$

The IB agent followed a softmax selection rule Eq (16). After an action is taken, likelihood $P(R|\hat{\theta}_n)$ is updated depending on a reward $R = 1$ or a no reward $R = 0$ is observed by the IB agent.

$$P(R = 1|\hat{\theta}_n(i))_n = \delta_i\hat{P}_n(R),\tag{4}$$

$$P(R = 0|\hat{\theta}_n(i)) = \delta_i(1 - \hat{P}_n(R)),\tag{5}$$

$$P(\hat{\theta}_n(i)|R) \approx P(R|\hat{\theta}_n(i)) * P(\hat{\theta}_n(i))\tag{6}$$

The multiplication of the prior probability of estimated set reward probabilities $P(\hat{\theta}_n(i))$ with the likelihood $P(R|\hat{\theta}_n(i))$ and normalization of this product, so that its integral (sum) equals to one, leads to the posterior probability of estimated set reward probabilities $P(\hat{\theta}_n(i)|R))$. The newly estimated set reward probabilities are sampled randomly from the range of posterior probabilities to update the previous estimates. The posterior probabilities become then the prior probabilities for the next trial. A noise factor of +/−2.5% or $\epsilon_n$, drawn from a uniform probability distribution with boundaries [-0.025,0.025], was added to the new estimates of the set reward probabilities $\hat{\theta}_n(i)$. This noise factor helps the model to explore wider range of possible set reward probabilities and converge more precisely to the true set probabilities. Since the noise term $\epsilon_n$ is added to the new estimates of the set reward probabilities, it is possible that some of these new estimates $\hat{\theta}_n(i)$ fall outside of the original boundaries of 0 and 1. We corrected for this as follows:

$$\begin{cases} \hat{\theta}_n(i) = 1, & if\ \hat{\theta}_n(i) > 1 \\ \hat{\theta}_n(i) = 0, & if\ \hat{\theta}_n(i) < 0 \end{cases}\tag{7}$$

A new iteration starts with trial $t = t + 1$ until the session is over at the end of the last trial $t = T$.

**LK model.** Different from standard RL models, the LK (name is after first and last authour of this paper) model performs two approximations concurrently that are interdependent. One approximation attempts to estimate the set reward probabilities, while the other attempts to approximate the "baiting state" of the environment. By combining these two, LK model approximates the actual reward probabilities.

The reward probability observed by an agent at trial $t$ of a given option $i$ in a VI schedule can be defined as

$$P(R|i, t + 1) = 1 - (1 - P_{set}(R|i, t))^{T_i+1}.\tag{8}$$

Where the $P_{set}(R|i, t)$ corresponds to the set reward probability of the option i at trial t, and $T_i$ is the number of trials since the option i was chosen. For cases where $P_{set}(R|i, t)$ is not equal to

$P_{set}(R|i, t-1)$ with $t > 1$ and $T_i > 0$, the reward probability can be described as

$$P(R|i, t+1) = 1 - (1 - P_{set}(R|i, t))(1 - P(R|i, t)\tilde{\delta}_{i,t}). \tag{9}$$

Where the term $\tilde{\delta}_{i,t} = 1$ when the option i at trial t was not selected, and 0 otherwise. Thus, when an option is selected, the immediate reward probability will follow its set reward probability. On the contrary, when it is not selected, it will increase its reward probability if the set reward probability $P(R|i, t) > 0$. The recursivity of Eq (9) follows a Markovian property, and hence, it is feasible to model it with a reinforcement learning (RL) model. In order for the RL model to estimate the reward probabilities in the task, it is necessary that it estimates the set reward probability. This model can estimate the set reward probability by using a Rescorla-Wagner rule

$$\hat{P}_{set}(R|i, t) = Q_{set}(i, t+1) = Q_{set}(i, t) + \alpha(R(t) - Q_{set}(i, t)), \tag{10}$$

when the choice of option i at t was the same as in t-1. This condition, however, would violate the Markovian property since it needs to keep track of the choices at t-1 and t, to estimate the upcoming reward probabilities. It is still a reasonable estimation if we remove such a condition. In addition, and for the sake of simplicity, the $Q$ replaces the estimation of reward probabilities $\hat{P}(R)$ notation. With $\alpha \in [0, 1]$ and, following Eqs (68) and (69), we can estimate the actual reward probability observed by our agent in an update equation as

$$Q(i, t+1) = 1 - (1 - Q_{set}(i, t+1))(1 - Q(i, t)\tilde{\delta}_{i,t}). \tag{11}$$

While this model is sufficient to solve a VI task, it is possible to modify it to make it adaptable to a wider number of tasks. In particular, the LK model can be adaptable to tasks where there is no baiting structure, such as in the armed bandit tasks. It can even incorporate a continuous variable that defines the influence of baiting on the upcoming reward probability. To control for these variants, a dynamic factor $\psi(t) \in [0, 1]$ can replace the discreet value $\tilde{\delta}_{i,t}$ and be included in the update equation

$$Q(i, t+1) = 1 - (1 - Q_{set}(i, t+1))(1 - Q(i, t)\psi(i, t+1)). \tag{12}$$

The dynamic factor $\psi(t)$ will have a value of 1 when the task is baited and 0 when it is not. It can be updated at trial t as follows

$$\psi(i, t+1) = \tilde{\delta}_{i,t}(\psi(i, t) + \alpha_\psi(t)(1 - \psi(i, t))). \tag{13}$$

The learning rate $\alpha_\psi(t)$ would define how "fast" the dynamic factor $\psi(t)$ will reach 1, and therefore, how "baited" the task is. The learning rate $\alpha_\psi(t)$ should then be affected by the prediction error of the baiting rate. Thus, a proposed update rule for this learning rate would follow the prediction error $PE(t) = R(t) - Q(i, t)$ and be computed as follows

$$\alpha_\psi(t) = \alpha_\psi(t-1) + \tau_\psi \left( \frac{PE(t) + 1}{2} - \alpha_\psi(t-1) \right). \tag{14}$$

Where $\tau_\psi \in [0, 1]$ is a fixed learning rate. A condition for this algorithm is that the rewards $R(t)$ are bounded between 0 and 1. This will ensure that $\frac{PE(t)+1}{2} \in [0, 1]$ and therefore $\alpha_\psi(t) \in [0, 1]$.

## Linear regression for reward and choice history effects

The influence of the past rewards $R(t - n)$ and choices $C(t - n)$, with $n = 1, 2, \ldots, N$ trials, on the upcoming choice $C(t)$ of an agent was represented by the coefficients $\hat{\beta}_{log-odds}$ in the linear domain of a logistic regression, also known as log odds. To calculate the regression coefficients, we defined the reward vector as the difference in experienced rewards between options as $R(t) = R_{Right}(t) - R_{Left}(t)$ at every trial $t$. The rewards take values $R_{Right}(t) = 1$ when the reward was delivered in that trial or $R_{Right}(t) = 0$ in all other cases, when subjects chose the right option and did not received a reward, or if the left option was chosen. The equivalent was true for $R_{Left}(t)$.

The choice vector was defined as $C(t) = 1$ if the right option was chosen at trial $t$ and $C(t) = 0$ otherwise. The logistic regression was then regularised with an elastic net that linearly combines penalties $L^1$ and $L^2$ from lasso and ridge regression [14]. The penalty term of the elastic net [15] is defined as $P(\beta_{log-odds})$, where $P$ is the number of parameters (number of coefficients-$\beta_{log-odds}$) and λ is the tuning parameter of this penalty term. The tuning parameter λ was selected after observing a minimum on the deviance in a cross-validation process with different λ values. The penalty term interpolates between the $L^1$ and $L^2$ norms as follows:

$$P(\beta_{log-odds}) = \lambda \sum_{j=1}^{p} (0.25 \beta_{log-odds,j}^2 + 0.5 |\beta_{log-odds,j}|). \tag{15}$$

In simple words, the regression coefficients that had the minimum mean deviance (or, equivalently, the maximum mean log-likelihood) as a function of the tuning parameter λ in a five-fold cross-validation process were selected as the estimated coefficients $\hat{\beta}_{log-odds}$ for past rewards and choices influencing the upcoming choice. We implemented the regularised logistic regression in MATLAB (www.mathworks.com) using the *lassoglm* function.

**Extended linear regression model.** Extended regression model includes effects of unsigned rewards $R(t - n) = R_{Right}(t - n) + R_{Left}(t - n)$, right rewards $R_{Right}(t - n)$, left rewards $R_{Left}(t - n)$, right no rewards $NR_{Right}(t - n)$, left no rewards $NR_{Left}(t - n)$ and choices $C(t - n)$ for past $n$ trials on the upcoming choice $C(t)$. The right no rewards $NR_{Right}(t - n)$ were defined as 1 when the right side was chosen and no reward observed at time $t$, and 0 otherwise. $NR_{Left}(t - n)$ follows the same definition for the left side respectively. The extended regression model followed the same elastic net regularization for parameter selection as described above.

## Optimisation of parameters for behavioural prediction

The choice history $\delta_i(1, 2, \ldots, t - 1)$ and reward history $R(1, 2, \ldots, t - 1)$, were introduced to various RL models as $x_{t-1}$ to compute the action values and determine the probability of choosing an action in the current trial $t$ using the softmax action selection rule:

$$P(a_t = i) = \frac{e^{\beta(value_{i,t})}}{\sum_{k=1}^{A} e^{\beta(value_{k,t})}}, \tag{16}$$

where $value_{i,t}$ is determined by the particular RL model (see more details in the description of RL models). Note that the softmax selection rule is not limited to two options, but the subsequent description of the optimization steps are limited to two option setups.

This probability $P(a_t = i)$ and the actual action $\delta_{i,t}$ at time $t$ with the set of parameters $\theta_{RL}(n)$ per model determine the log-likelihood as follows:

$$l_t(n)(\theta_{RL}(n)|x_{t-1}) = \delta_{i,t} ln(P(a_t = i|\theta_{RL}(n))) + (1 - \delta_{i,t}) ln(1 - P(a_t = i|\theta_{RL}(n))). \tag{17}$$

To find the optimal parameters $\hat{\theta}_{RL}$ for each model, we initially selected 1,000 combinations $\theta_{RL,1}$ from a uniform distribution, where the boundaries of the parameter space were $\alpha \in [0, 1]$, $\beta \in [0, 50]$, $\varphi \in [-25, 25]$, $\vartheta \in [-25, 25]$, $\iota \in [-25, 25]$, $\tau_F \in [0, 1]$, $\tau_M \in [0, 1]$ and $\tau_S \in [0, 1]$. Next, we took 1% of the combinations with the maximum mean log-likelihood and used the mean $\bar{\theta}_{RL,1,1\%}$ and standard deviation $\sigma(\theta_{RL,1,1\%})$ of this subset to draw a new set of 1,000 combinations as follows:

$$\theta_{RL,2}(n) \sim \mathcal{N}(\bar{\theta}_{RL,1,1\%}, \sigma(\theta_{RL,1,1\%})). \tag{18}$$

We repeated this process several times to narrow the original parameter space until the highest log-likelihood (a negative value or zero) was less than 99.99% of its value for two iterations in a row. The optimal parameters for the prediction of each model $\hat{\theta}_{RL}$ were then the combination of parameters with the highest mean log-likelihood of the last iteration in the optimisation process. This optimization resembles the simulated annealing algorithm [16].

The optimised parameters for each model and the estimated coefficients were trained and tested via five-fold cross-validation on the behavioural data in order to obtain the average of the minimum negative log-likelihood and the average AUC. These latter metrics were used for model selection and the goodness of prediction for each model, respectively. To compute the AUC, the probability $P(a_t = i|\hat{\theta}_{RL})$ was set as the score and the original action $\delta_{i,t}$ as the label. We used MATLAB function *perfcurve* to compute the AUC scores.

## Optimisation of parameters in simulated environments

Regret Eq (72) was used to estimate the optimal parameters for each model in the VI task. We iteratively determined the best parameters for each model by selecting a random set of initial parameters $\theta_1$ uniformly 10,000 times and by simulating one session of $T = 1000$ trials four times. We tested two conditions, the first one was the VI task under different degrees of volatility (i.e. block sizes): $N_{bl} \in \{50, 100, 500\}$ with two pair of the set reward probabilities 0.10 vs. 0.40 and 0.4 vs. 0.1 for left and right sides respectively. The second condition of the VI task have three different pairs of set probabilities 0.20 vs. 0.80, 0.30 vs.0.70 and 0.40 vs. 0.60, and one block size $N_{bl} = 100$. The boundaries of the parameter space were $\alpha \in [0, 1]$, $\beta \in [0, 10]$, $\varphi \in [-2, 0]$, $\vartheta \in [0, 2]$, $\tau_F \in [0, 1]$ and $\tau_S \in [0, 1]$. Since the number of combinations was insufficient for a thorough search of the parameter space, we selected the 5% with the lowest regret $(\hat{\theta}_1)$ out of the original 1,000 combinations.

We thereby realised a local search algorithm in our multidimensional parameter space, extending three standard deviations around the top 5% of the previous parameter combinations and selecting a new random set of parameters $\theta_2$, 5000 times:

$$\theta_2(n) \sim \mathcal{N}(\hat{\theta}_1, 3\sigma(\hat{\theta}_1)). \tag{19}$$

We repeated the selection of best parameters by for $\theta_2$ the same way described for $\theta_1$. It is important to mention that at every iteration, a different random number generator was selected to prevent running the generator under the same probabilities during each trial and to give the model the opportunity to explore different probabilistic scenarios. However, this did not guarantee that the combinations had low regret due to this random selection. To overcome this potential problem, we took the 10% of $\theta_2$ with the lowest regret, having $\hat{\theta}_2$, and tested each combination 100 times under different scenarios to determine consistency of the same outcomes. The parameter combination in $\hat{\theta}_2$ with the lowest regret on average was selected as optimal.

## From logistic regression to DT model

**Logistic regression model of behaviour.** The probability that a foraging animal will choose the left side in a discrete version of a VI task with two choices is $P(a_t = l)$. Lau and Glimcher [13] originally proposed to compute the probability $P(a_t = l)$ as a linear combination $h_{l,t}$ of the reward $R$ and choice $C$ history of the last $M$ trials with their corresponding coefficients $b_R$, $b_C$, plus a constant bias $K$. As the choices are binary in nature, the linear function $h(t)$ is then transformed in logit space to a probability, as shown in Eq (20). Here, $\beta$ is the exploration factor. This is also known as a log-linear model or logistic regression:

$$P_h(a_t = l) = \frac{1}{1 + e^{-\beta h_{l,t}}} \tag{20}$$

and

$$h_{l,t} = \sum_{m=1}^{M} b_{R,m} \delta_{l,t-m} R_{t-m} - \sum_{m=1}^{M} b_{R,m} \delta_{r,t-m} R_{t-m} + 2\sum_{m=1}^{M} b_{C,m} \delta_{l,t-m} + K, \tag{21}$$

where the chosen action $i$ at time $t$ is depicted as $\delta_{i,t} = 1$ and the reward at time $t$ is described as $R_t = 1$. $K$ is the bias term in the equation.

**Decay in the reward and choice history regression coefficients.** The results of our experiments and in previous studies [2, 13, 17] show that $b_{R,m}$ and $b_{C,m}$ decay, as a function of the m trial history. We propose that these decays for the reward and choice history to be modelled according to the following equation:

$$b_{R,m} = \alpha(1 - \alpha)^{m-1}. \tag{22}$$

Here, $\alpha$ characterises the decay rate for rewards. $\tau_F$ and $\tau_S$ characterise the decay rate for choices according to:

$$b_{C,m} = \varphi(1 - \tau_F)\tau_F^{m-1} + \vartheta(1 - \tau_S)\tau_S^{m-1}. \tag{23}$$

where $\varphi$ is the scaling factor for choice trace $F$, and $\vartheta$ is the scaling factor for choice trace $S$.

The function $h_t$ of the logistic regression was separated into reward and choice terms to ensure simplicity in the next analysis.

$$h_{l,t} = h_{R,t} + h_{C,t} + K, \tag{24}$$

$$h_{l,R,t} = \sum_{m=1}^{M} \alpha(1 - \alpha)^{m-1} \delta_{l,t-m} R_{t-m} - \sum_{m=1}^{M} \alpha(1 - \alpha)^{m-1} \delta_{r,t-m} R_{t-m}, \tag{25}$$

and

$$h_{l,C,t} = 2\sum_{m=1}^{M} (\varphi(1 - \tau_F)\tau_F^{m-1} + \vartheta(1 - \tau_S)\tau_S^{m-1}) \delta_{l,t-m}. \tag{26}$$

In the next section, we show how these equations were used to derive the recursive DT update rule:

**From logistic regression to the softmax selection rule.** Like the logistic regression model Eq (20), the RL model computes choice probabilities using the value expectation of rewards (Q, Eq (69) under the softmax selection rule as follows:

$$P_{RL}(a_t = l) = \frac{1}{1 + e^{-\beta(Q_{l,t} - Q_{r,t})}}. \tag{27}$$

$\beta$ defines the inverse temperature (also called the exploration factor), and $Q_{i,t}$ is the reward expectation value of a given action (or option) $i$ at time $t$. In this case, $l$ and $r$ refer to the left and right options, respectively. We propose that one way to reconstruct the animals' choice history effects within the RL framework is to incorporate a separate double-choice history component in the softmax selection rule. Using the difference between the two choice traces, each computed with a different learning rate, we can recover the characteristic (wavy) shape of the animals' choice history effects. Here, we use learning rates that are independent, such as $\tau_F$ and $\tau_S$, and which can generate a vast diversity of curves. The softmax function then takes the following form:

$$P_{RL}(a_t = l) = \frac{1}{1 + e^{-\beta(Q_{l,t} - Q_{r,t} + \varphi F_{l,t} - \varphi F_{r,t} + \vartheta S_{l,t} - \vartheta S_{r,t})}}. \tag{28}$$

Here, $F$ is the fast decaying choice trace, and $S$ is the slowly decaying choice trace. These traces are described in detail by Eqs (70) and (71).

Next, we show the equivalence of the Eq (28) with the logistic regression model (Eq (20)). Therefore, the following equivalence should be satisfied:

$$P_h(a_t = l) = P_{RL}(a_t = l). \tag{29}$$

It follows from this equivalence that

$$h_{l,t} = Q_{l,t} - Q_{r,t} + \varphi F_{l,t} - \varphi F_{r,t} + \vartheta S_{l,t} - \vartheta S_{r,t}. \tag{30}$$

**Reward expectation and its equivalence in logistic regression.** For the reward expectation, it was already shown that the logistic regression model is equivalent to the F-Q model [18], which is characterised by the following update rule:

$$Q_{l,t} = Q_{l,t-1} + \alpha(\delta_{l,t-1} R_{t-1} - Q_{l,t-1}), \tag{31}$$

where $\alpha$ is the learning rate. Katahira [18] proposed that the equivalence in Eq (31) can be reduced to Eq (32):

$$Q_{l,t} = \left(\prod_{m=1}^{t-1}(1-\alpha)^m\right) Q_{l,1} + \alpha \sum_{m=1}^{t-1}(1-\alpha)^{m-1} \delta_{l,t-m} R_{t-m}. \tag{32}$$

There are enough trials $t >> 1$ to neglect events that occurred in the very distant past [18]:

$$\left(\prod_{m=1}^{t-1}(1-\alpha)^m\right) Q_{i,1} \approx 0. \tag{33}$$

Under this assumption, we define $t - 1 \approx M$, and, therefore, it is possible to observe how this update rule for the reward expectation is equivalent to the reward coefficients in logistic regression:

$$Q_{l,t} - Q_{r,t} = \alpha \sum_{m=1}^{M}(1-\alpha)^{m-1} \delta_{l,t-m} R_{t-m} - \alpha \sum_{m=1}^{M}(1-\alpha)^{m-1} \delta_{r,t-m} R_{t-m}. \tag{34}$$

It follows that,

$$h_{l,R,t} = Q_{l,t} - Q_{r,t}. \tag{35}$$

**Choice traces and their equivalences in logistic regression.** In the next section, we demonstrate how to reconstruct the proposed choice history decay observed in the experimental data in a recursive manner:

$$h_{l,C,t} = \varphi F_{l,t} - \varphi F_{r,t} + \vartheta S_{l,t} - \vartheta S_{r,t} + K. \tag{36}$$

We introduce a recursive function that we call a choice trace. Its equivalence in logistic regression is shown in the next three equations. We follow similar derivations as shown by Katahira in his work [18]. We introduce the functions fast (F) and slow (S), corresponding to the fast and slow learning rates in the update rules, respectively (Eqs (70) and (71)).

$$F_{l,t} = (1 - \tau_F)F_{l,t-1} + \tau_F \delta_{l,t-1} \tag{37}$$

We show the equivalence of the choice trace with the logistic regression in the following derivation:

$$F_{l,t} = \left(\prod_{m=1}^{t-1}(1 - \tau_F)^m\right)F_{l,1} + \tau_F \sum_{m=1}^{t-1}(1 - \tau_F)^{m-1}\delta_{l,t-m}. \tag{38}$$

Under the same assumption as that of the reward expectation, when the number of trials is high enough $t >> 1$ that very distant past events become negligible, we can reduce the previous equation as follows:

$$F_{l,t} = \tau_F \sum_{m=1}^{t-1}(1 - \tau_F)^{m-1}\delta_{l,t-m}. \tag{39}$$

In the next equations, we can observe how the choice trace of the chosen side is the same as that minus the choice trace of the unchosen side in a two-alternative choice task.

$$\delta_{r,t} = 1 - \delta_{l,t} \tag{40}$$

$$F_{r,t} = \left(\prod_{m=1}^{t-1}(1 - \tau_F)^m\right)F_{r,1} - \tau_F \sum_{m=1}^{t-1}(1 - \tau_F)^{m-1}\delta_{l,t-m} + \tau_F \sum_{m=1}^{t-1}(1 - \tau_F)^{m-1} \tag{41}$$

When $t >> 1$,

$$\tau_F \sum_{m=1}^{t-1}(1 - \tau_F)^{m-1} = 1. \tag{42}$$

Finally,

$$F_{r,t} = 1 - F_{l,t}. \tag{43}$$

Therefore, the difference between the two choice traces with the same learning rate can be represented as follows:

$$F_{l,t} - F_{r,t} = F_{l,t} + F_{l,t} - 1 = 2F_{l,t} - 1 \tag{44}$$

and

$$F_{l,t} - F_{r,t} = 2\tau_F \sum_{m=1}^{M}(1 - \tau_F)^{m-1}\delta_{l,t-m} - 1. \tag{45}$$

The second choice trace is called slow because the learning rate is slower than that in the

previous choice trace by definition. The equivalence of this recursive equation with a sum and a decay follows the same logic as the previous choice trace.

$$S_{l,t} = (1 - \tau_S)S_{l,t-1} + \tau_S\delta_{l,t-1}, \tag{46}$$

$$S_{l,t} = \left(\prod_{m=1}^{t-1}(1 - \tau_S)^m\right)S_{l,1} + \tau_S\sum_{m=1}^{t-1}(1 - \tau_S)^{m-1}\delta_{l,t-m}. \tag{47}$$

Assuming that $t \gg 1$,

$$\tau_S\sum_{m=1}^{t-1}(1 - \tau_S)^{m-1} = 1, \tag{48}$$

$$S_{r,t} = 1 - S_{l,t}, \tag{49}$$

and

$$S_{l,t} - S_{r,t} = 2\tau_S\sum_{m=1}^{M}(1 - \tau_S)^{m-1}\delta_{l,t-m} - 1. \tag{50}$$

Based on the choice part of Eq (30), we obtain the following equations:

$$\varphi F_{l,t} - \varphi F_{r,t} + \vartheta S_{l,t} - \vartheta S_{r,t} = 2(\varphi F_{l,t} + \vartheta S_{l,t}) - \varphi - \vartheta, \tag{51}$$

$$2(\varphi F_{l,t} + \vartheta S_{l,t}) - \varphi - \vartheta$$
$$= 2(\varphi\sum_{m=1}^{M}\tau_F(1 - \tau_F)^{m-1}\delta_{l,t-m} + \vartheta\sum_{m=1}^{M}\tau_S(1 - \tau_S)^{m-1}\delta_{l,t-m}) - \varphi - \vartheta, \tag{52}$$

and

$$K = \varphi + \vartheta. \tag{53}$$

We then show how, using the fast and slow choice traces, we can recover Eq (30) as a part of the original logistic regression, where $\varphi$ and $\vartheta$ are the scaling factors. These factors allow the agents to select which choice trace, fast or slow, should dominate the decisions.

$$2(\varphi F_{l,t} + \vartheta S_{l,t}) = 2\sum_{m=1}^{M}(\varphi\tau_F(1 - \tau_F)^{m-1} + \vartheta\tau_S(1 - \tau_S)^{m-1})\delta_{l,t-m} = h_{l,Ch,t} \tag{54}$$

Thus,

$$\varphi F_{l,t} - \varphi F_{r,t} + \vartheta S_{l,t} - \vartheta S_{r,t} = h_{l,Ch,t} + K. \tag{55}$$

## RL models

**Indirect actor model.**   The indirect actor model updates the reward expectation (or state-action value) Q only for the chosen option.

$$Q_{i,t} = Q_{i,t-1} + \delta_{i,t-1}\alpha(R_{t-1} - Q_{i,t-1}), \tag{56}$$

$$value_{i,t} = Q_{i,t}. \tag{57}$$

**Direct actor model.**   The reward expectation of the direct actor model is updated based on the probability of the chosen action and the reward outcome. This rule also affects the reward expectation value of the unchosen actions.

$$Q_{i,t} = Q_{i,t-1} + \alpha(\delta_{i,t-1} - P(i_{t-1}))(R_{t-1} - c), \tag{58}$$

$$value_{i,t} = Q_{i,t}. \tag{59}$$

Here c is a parameter that we fit to the behavioral data and can be seen as average reward rate in R-learning models [19]. The direct actor model is equivalent to the actor critic model with one state [20]. Since it directly updates the choice probability or policy $P(i_t)$, it can be considered as a policy update model [21]. The policy update also happens in the actor critic model where the actor is using reward prediction errors to update its policy.

**F-Q down model.**   The F-Q model is a slight modification of the indirect actor model, where the reward expectation value of the unchosen actions are forgotten and vanish to zero.

$$Q_{i,t} = Q_{i,t-1} + \alpha(\delta_{i,t-1}R_{t-1} - Q_{i,t-1}), \tag{60}$$

$$value_{i,t} = Q_{i,t}. \tag{61}$$

**F-Q with the choice trace model.**   According to the F-Q W/C model, the probability of taking the next action will depend not only on the reward expectation but also on choice trace F updated according to the following:

$$Q_{i,t} = Q_{i,t-1} + \alpha(\delta_{i,t-1}R_{t-1} - Q_{i,t-1}), \tag{62}$$

$$F_{i,t} = F_{i,t-1} + \tau_F(\delta_{i,t-1} - F_{i,t-1}), \tag{63}$$

$$value_{i,t} = Q_{i,t} + \varphi F_{i,t}. \tag{64}$$

**F-Q up model.**   While the F-Q model decreases the value of the unchosen actions, the proposed model increases their value up to $C$ with its own learning rate $\alpha_{up}$, which acts as a positive counter for unchosen actions that adds to the action value.

$$Q_{i,t} = Q_{i,t-1} + \delta_{i,t-1}\alpha(R_{t-1} - Q_{i,t-1}) + (1 - \delta_{i,t-1})\alpha_{up}(C - Q_{i,t-1}), \tag{65}$$

$$value_{i,t} = Q_{i,t}. \tag{66}$$

The initial value of $Q_{i,1}$ is set from a uniform random distribution constrained between [0, 1].

### Notations and symbols

All notations and symbols are provided in S1 Appendix.

### Participant instructions

The following text is the instruction (verbatim) given to the subjects who participated in human-VI task: *"In this experiment you will play a game about collecting apples. The game takes around 20–30 min to complete. When playing this game please choose a place with little distractors around for the time of the game. Please play until a notification tells you that you have completed enough rounds. After that you may continue playing or quit the game. To quit, use the "escape" key. After you have finished all rounds, there will be a short questionnaire. If you wish to end the game before you have completed all rounds, use the "escape" key. You will not be directed to the questionnaire and your data will not be used. In the game you play an avatar who collects apples from two identical-looking trees. To begin the game, you need to hold down the "space" key until the gate in front of the two trees opens. After the avatar has stopped moving, use the arrow keys to choose between the trees. A falling apple will visualize whether you received an apple in that round."*

### Implementation

The analyses of the behavioural and computational models were performed using MATLAB (MathWorks, Inc.).

## Results

To understand the adaptive function of choice history effects in animals, we first tested if they would emerge in optimal models of behavior. To do this, we constructed several optimal agents that maximize reward rates in the VI task. This task is a variant of the two-alternative forced-choice (2AFC) task with a discrete version of VI schedule of reinforcements [13]. Henceforth, we refer to this task simply as the VI task. In this task, the rewards of two alternative options are delivered probabilistically following a baiting schedule (Fig 1A and 1B). The reward probabilities in the VI task that we used to test all of the agents (mice, humans, and synthetic) were updated (baited) in each trial according to the Eq (1), which we also reiterate here:

$$P(R|i, t + 1) = 1 - (1 - P_{set}(R|i, t))(1 - P(R|i, t)(1 - \delta_{i,t})). \tag{67}$$

Where $\delta_{i,t} = 1$, when an agent chooses option $i$ at time $t$; otherwise, $\delta_{i,t} = 0$. Here, $P(R|i,t)$ represents the probability of receiving reward $R$ at trial $t$ for action $i$ and is updated for each trial. Different from $P(R|i,t)$, the set reward probability $P_{set}(R|i,t)$ is changing only during block transitions and is independent of an agent's choices (Fig 1A). According to Eq (67) reward probabilities for different set reward probabilities grow at different rates (Fig 1B).

We studied the performance of several optimal agents, increasing in their biological plausibility: 1. The Oracle agent which has full knowledge of the objective reward probabilities and task structure (Fig 1C, left panel). 2. The inference-based (IB) agent that uses elements of Bayesian formalism to infer the reward probabilities based on outcome experiences. This agent updates its beliefs based on the VI schedule of rewards (Fig 1D, Eq (67)). 3. The RL agent (LK model) estimates the baiting rate of reward probabilities and uses a Markovian framework to update its action values following a VI schedule of rewards (Fig 1E, left panel). These agents are described fully in the Materials and methods section regarding optimal agents.

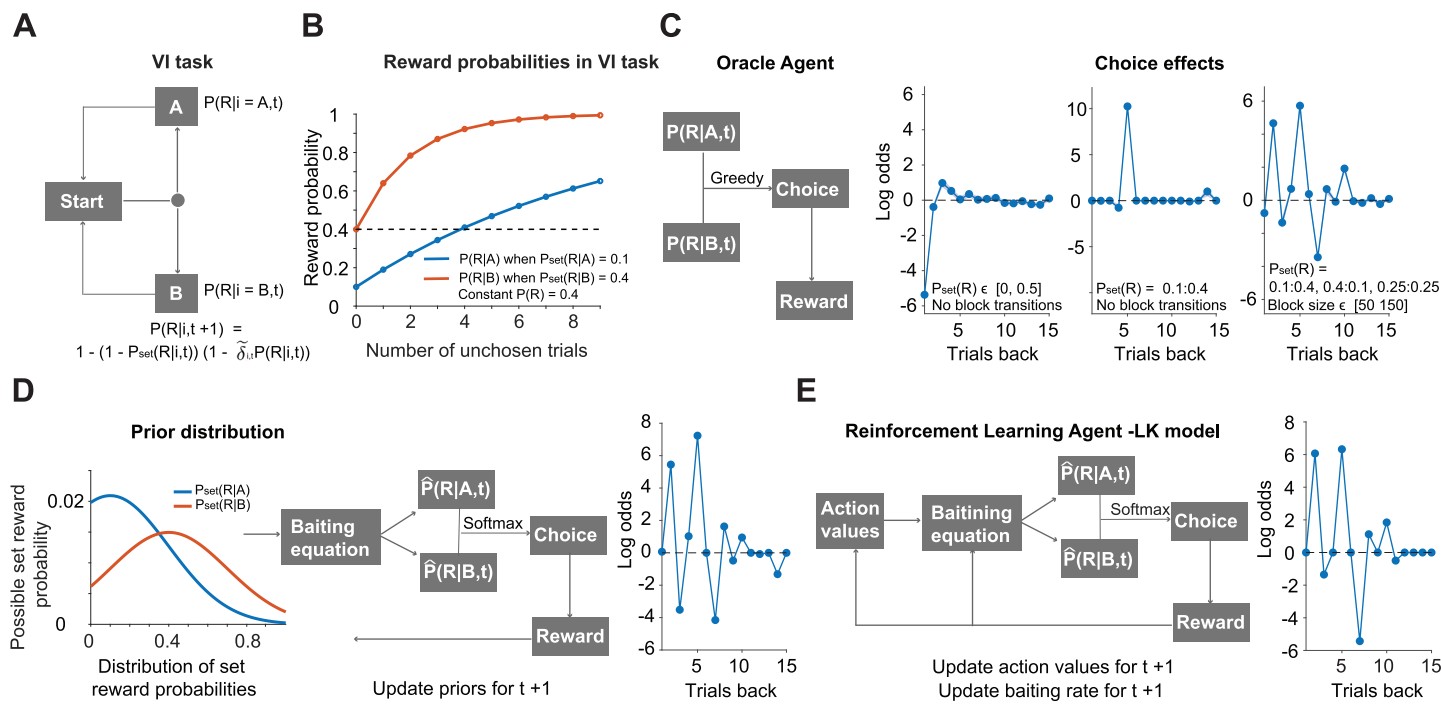

**Fig 1. Optimal models and their choice history effects in the VI task.** (A) Schematic of the VI task and the equation for updating the reward probabilities of each option i (A vs. B or left vs. right). Once the agent initiates a trial at t, it can choose between options A or B. Each option has its own dynamic probability of reward delivery P(R). (B) Change in reward probability in the VI task is determined by the set reward probability and the update rule following the baiting equation depicted in (A). (C) In the left-hand panel, a scheme of how the Oracle Agent solves the VI task. The Oracle agent "knows" the exact reward probabilities for each option in each trial. It chooses the option with the highest probability of reward in a greedy manner. After making a choice, it observes a reward or no reward. The second panel shows the influence of past choices on current choice when the set reward probabilities of each option are sampled from a uniform distribution ∈ [0,0.50]. The third panel shows the influence of past choices on the current choice with the set reward probabilities of 0.1 vs. 0.4 for the left and right options, respectively. On the rightmost panel, we added to previous set reward probabilities 0.25 vs. 0.25 reward probability trials with block transitions as described in the main text. The x-axis depicts trials back in history and the y-axis logistic regression coefficients. (D) In the left-hand panel, we show how a IB agent solves the VI task. First, at trial t, the IB agent samples the estimated set reward probabilities for each option i from prior distributions. Then, it uses the baiting equation to estimate the reward probabilities for each option. Using a softmax action selection rule, it makes a decision. After observing a reward or the absence of a reward, it updates the priors for the next trial. The rightmost panel shows the influence of past choices on the IB agent's next decision, when the VI task had set reward probability pairs of 0.10 vs. 0.40, 0.4 vs. 0.1, and 0.25 vs. 0.25. The VI task included possible changes in the set reward probability pairs established at each block transition with a duration of 50 to 150 trials (randomly sampled from a uniform distribution) (E) The left panel shows how a reinforcement learning (RL) agent using a newly derived LK model solves the task. According to the LK model, the action values are converted into choices using the baiting update equation and the softmax selection rule. After observing a reward or the absence of a reward, the model updates the action values and the estimated baiting rate for the next trial. The right panel shows the influence of past choices on the LK model's next decision, when the the VI task had set reward probability pairs of 0.10 vs. 0.40, 0.4 vs. 0.1 and 0.25 vs. 0.25. We used the same block structure for the LK model as one used for the IB agent.

## Choice history effect in optimal agents

In optimal agents we focused our analysis only on the choice history effects as reward probabilities are "known" or quickly inferred by these agents and do not contribute significantly to current choices. To quantify the effect of past choices on current choices, we used the agent's past choices to predict its current choices. To do this, we analyzed the choice dynamics using a regularized regression analysis with a cross-validated elastic net (see linear regression in the Materials and methods section) [14].

We tested the performance of the Oracle agent using three different conditions. These conditions varied from more general to specific task conditions that we also used for mice and humans. 1. The set reward probabilities were drawn from a uniform distribution $P_{set}(R|i) \in$ [00.5], with $i \in A, B$ for possible options and remained constant within a given session. Each session consisted of T = 1000 trials, and we used a total of n = 400 sessions. 2. The set reward probability consisted of pairs of 0.10:0.40 and 0.40:0.10, where each set reward probability pair

was assigned randomly for each session. Each session consisted of T = 1000 trials and we used a total of n = 10 sessions. 3. The set reward probability pairs of 0.10:0.40, 0.40:0.10, and 0.25:0.25 per block, were randomly assigned to each block. Each block consisted of trials of T ∈ [50, 150] length, drawn from a uniform distribution, and a virtual session contained 9 blocks. Therefore, the total length of trials was T ∈ [450, 1350]. The Oracle agent was run on each condition for 10 sessions. This task condition was the closest to that used to test the mice and humans in a VI task, and it was thus further used to test other optimal models.

For the IB agent, we used n = 100 sessions and for the LK model we used n = 5000 sessions. The influence of past rewards and past choices for all optimal agents was analyzed using regression with LASSO regularization. The coefficients with the lowest deviation were selected in a five-fold cross-validation process (see Materials and methods section for details). For the action selection, we used a greedy algorithm for the Oracle agent and the softmax selection rule Eq (68) for the IB and LK agents. This was justified by the fact that the Oracle agent has full access to the exact reward probabilities of options and does not need to explore better options, whereas the LK and IB agents need to learn over the course of numerous trials which options provide higher reward rates.

For the Oracle agent, the regression analysis revealed a strong alternation effect for immediate past choices, and perseverance effect for choices further back in the past (Fig 1C, second panel from left) when the set reward probabilities were drawn from a uniform distribution with no block transitions (condition 1 for the oracle agent, see above). Here, we define alternation as a lower probability of repeating the same choices as opposed to making different choices and perseverance as a higher probability of repeating the same choices as opposed to making different choices. Thus, the short-term alternation and long-term perseverance of choices simply reflect how set reward probabilities drawn from uniform distribution are updated in the VI task (S1(A) Fig). It was not feasible to test animal performance using uniform set reward probability distributions to match the task conditions for the Oracle agent. This would require us to test each animal on a minimum of 400 sessions with it's unique set reward probability drawn from a uniform distribution. Therefore, we also used a smaller number of set reward probabilities for the Oracle agent that were later used to test choice history effects in animals. By analyzing the behavior of the Oracle agent, we found sharp peaks in choice history effects (Fig 1C, middle and right panels showing choice effects) that were unlikely to be seen in animals. These peaks (also seen in the IB and LK models) must arise from optimal agents inferring the exact set reward probabilities. This is particularly evident for set reward probabilities of 0.1 versus 0.4 without any block transitions with the Oracle agent (Fig 1C, middle panel showing choice effects). The Oracle agent repeated the same choice for the low probability (0.1) option in every fifth trial. Therefore, a sharp peak in the regression coefficient appears at exactly five trials back. When a VI task consists of block transitions with different set reward probabilities, these peaks appear at multiple trials back, potentially reflecting different choice sequences.

Very similar choice history effects were observed in the IB agent (Fig 1D, rightmost panel). The beliefs of the IB agent were characterized by a flat prior distribution over reward probabilities $P(R|i) \in [0\ 1]$ for both options, and the agent updated these beliefs for the chosen and unchosen options following the baiting schedule (Eq (67)) and Bayesian inference (Fig 1D, left and middle panels, also see the IB agent in the Materials and methods section). The choice history effects of the IB agent were estimated by the same regression analysis as used for the Oracle agent. We also verified that the IB agent achieved near optimal reward harvesting efficiency (S1(B) Fig) and also correctly inferred the set reward probabilities (S1(C) Fig).

Finally, we tested the RL agent (LK model), which learns the action values of the two options by estimating the baiting rate (LK model in Materials and methods) updated with the baiting schedule according to the Eq (67) (Fig 1E, left panel). The choice history effects of the

LK model closely resembled the other optimal agents performances and achieved near optimal reward harvesting efficiency in the simulated environment (S1(B) Fig). The choice history effects of the LK model were estimated by the regression analysis, as for the IB and Oracle agents (Linear regression for reward and choice history effects in Materials and methods).

## Short-term alternation and long-term perseverance of choices in mice and humans

Next, we sought to elucidate choice history effects in mice and humans performing the VI task. In contrast to the optimal agents, set reward probabilities as well as the baiting structure of the task were hidden from the mice and humans. We reasoned that mice and humans could estimate reward probabilities based on their reward history. Therefore, we anticipated that our test subjects may need to track of the history of rewards and choices. The subjects in the VI task had to choose either left- or right-side option after initiating the trial by pressing the space bar (Fig 2A) for humans or poking with the nose into the center port (Fig 2B) for mice. The

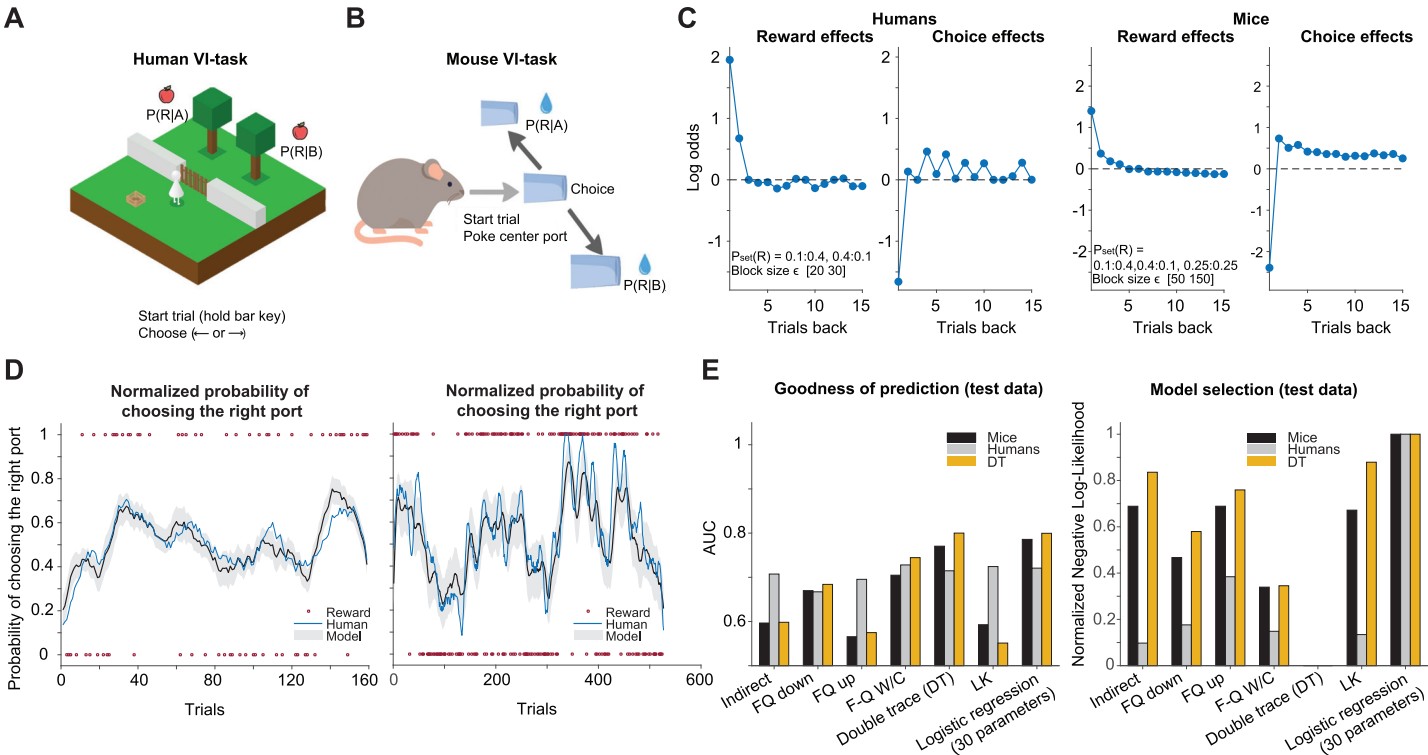

**Fig 2. Choice and reward history effects in test subjects and performance of the DT model in a VI task.** (A) Snapshot of the VI task in a computer game played by human participants. After opening a virtual fence by pressing the space bar on the keyboard, subjects had to wait from 0 to 5 seconds before making the decision to press the left or right key. The rewarded trials were indicated by a collection of virtual apples. (B) The scheme of the VI task adapted for water-deprived mice. The rodents had to poke the center port to start a trial and wait at the center port from 0.2 to 0.4s before choosing the right or left port. In the case of rewarded trials, the mice received 2 μL of water. (C) The influence of past rewards and choices on current choice for rodents (219 sessions with 7 mice) and human subjects (25 sessions with 19 subjects) was analyzed by logistic regression with LASSO regularization. The coefficients with the lowest deviation were selected in a five-time cross-validation process. Videos are available for this figure as follows: S1 Video for Human VI task. S2 Video 2 for Mouse VI Task. (D) Sample sessions of the choices made by a mouse (left panel) and a human (right panel), together with the choices predicted by the DT model; the mean prediction of the DT model is represented in bold lines and the standard deviation as shadows. Rewards are represented as red circles with the value of 1 when the right option was rewarded and 0 when the left option was rewarded. There are no red circles when a trial was not rewarded. (E) In the left panel, we show the area under the receiver operating characteristics curve (AUC) in the test data to determine the goodness of prediction of the mouse and human behavioral data. In the right panel, the negative log-likelihood in the test data normalized between the models for model-selection to predict the behavioral data of mice and humans. The direct actor model results were omitted from the graphs because their negative log probability tended to infinity. The best fit parameters for mouse behavior using the DT model were $\alpha = 0.75$, $\beta = 1.60$, $\tau_F = 0.71$, $\tau_S = 0.24$, $\vartheta = 3.31$, and $\varphi = -2.22$. The parameters for human behavior were $\alpha = 0.73$, $\beta = 2.62$, $\tau_F = 0.56$, $\tau_S = 0.40$, $\vartheta = 1.97$, and $\varphi = -2.10$.

specific set reward probabilities for the left- versus right-hand side that we used for the VI task were, respectively, 0.25:0.25, 0.40:0.10, and 0.10:0.40 for mice (animals, n = 7) and 0.40:0.10 and 0.10:0.40 for humans (subjects, n = 19). Each block consisted of 50–150 trials for mice and 20–30 trials for humans. Each mouse completed 1–27 blocks per sessions, while human subjects completed at least 6 blocks per session. We used a different number of trials per session with only two set reward probabilities for the humans for the following reason. Humans can typically tolerate shorter tasks, before their attentional and motivational states diminish, and thus we were constrained to limit the number of trials in humans.

Logistic regression with elastic-net regularization (Linear regression for reward and choice history effects in Materials and methods) showed monotonic diminishing effects of past rewards on current choices. The effects of past choices showed a mixture of short-term alternation and long-term perseverance (Fig 2C and S2 Fig) similar to what was observed with the Oracle agent for uniform set reward probability distributions (Fig 1C, left panel for choice effects). The logistic regression analysis of a monkey's choice from previous work [13] also showed qualitatively similar effects of past rewards and choices on current choices. The discrepancy in choice history effects of optimal agents (Fig 1C–1E, see panels showing choice effects except one with uniform probability distribution) and animals (Fig 2C, see panel for choice effects) must stem from the fact that optimal agents can precisely infer reward probabilities, while animals cannot. This suggests that the lack of precise knowledge of set reward probabilities in animals is replaced by the simple choice heuristic that assumes uniform distributions of set reward probabilities (compare Fig 1C second left panel and Fig 2C, panel for choice history effects). The shape of choice history effects are recovered also when, we compute fractional difference between consecutive choices and alternation on last alternation trial in ideal agent (S1(A) Fig). This agent that like the Oracle agent always chooses the higher probability options. To run this analysis the set reward probabilities were sampled from uniform probability distribution.

One alternative interpretation of choice history effects is that they reflect some reward contingencies. We decomposed reward history effects into multiple components. Specifically, we sought to reveal the contribution of past rewards unlinked to the previous choice (unsigned rewards), the past reward contribution linked to past choice, and the contribution of past no rewards linked to past choices, to current choices (see extended linear regression model in Materials and methods for linear regression for reward and choice history effects). This analysis failed to reveal any effect of unsigned past rewards on choices, but did show a clear effect of past no rewards on current choices (S3 Fig). This result suggests but does not prove that short-term alternation in choice history effects arises from behavioral responses to no rewards. However a slow, long-term perseverance of choices is difficult to explain by any reward contingencies. From this analysis it was also clear that average unsigned rewards have no effect on current choices. Thus, choice history effects may reflect complex dependencies of choices on immediate past no rewards, (short-term alternation of choices) and on more distant past choices. In either case we refer to these effects simply as choice history effects.

## The double trace model incorporates choice history effects observed in animals into action selection

Based on our previous results (Fig 1E), it became clear that animals do not follow an LK model (an RL model that optimally updates reward probabilities) to solve the VI task. This led us to derive a new RL model inspired by animal behavior. For this, we parametrized the choice history effects observed in animals into a standard RL model (see Materials and methods for derivation of the DT model). The DT model captures the choice history effects by including the

sum of two scaled choice traces $F$ and $S$, hence the double trace name. The choice traces $F$ and $S$, thus represent short-term alternation and long-term perseverance of choices respectively. These traces were updated with independent learning rates (Eqs (4) and (5)) similar to the update rules for a single choice trace described in [22]. Reward history effects were captured using the value update Eq (69), similar to the forgetting Q model [23]. According to the DT model, the probability that an agent will choose action $i$ (i.e., the right- or left-hand action) at time step $t$ follows the softmax action selection rule:

$$P(a_t = i) = \frac{e^{\beta(Q_{i,t} + \varphi F_{i,t} + \vartheta S_{i,t})}}{\sum_{k=1}^{A} e^{\beta(Q_{k,t} + \varphi F_{k,t} + \vartheta S_{k,t})}}, \tag{68}$$

where $Q_{i,t}$ denotes reward expectation, $F_{i,t}$ is the fast choice trace, $S_{i,t}$ is the slow choice trace, and $\beta$ is the inverse temperature. The lower the inverse temperature, the more stochastic the choices are, and thus, this quantity can be thought of as an exploration factor. The scaling factors for the choice traces are $\varphi$ and $\vartheta$. Each choice trace has a learning rate of either $\tau_F$ or $\tau_S$. $F$ and $S$ choice traces extracted from animals typically decay at fast and slow rates, respectively (i.e., $\tau_F \geq 0.5$ and $\tau_S < 0.5$). The reward expectation and choice traces are updated with the learning rates of $\alpha$, $\tau_F$, and $\tau_S$ as follows:

$$Q_{i,t} = Q_{i,t-1} + \alpha(\delta_{i,t-1} R_{t-1} - Q_{i,t-1}), \tag{69}$$

$$F_{i,t} = F_{i,t-1} + \tau_F(\delta_{i,t-1} - F_{i,t-1}), \ and \tag{70}$$

$$S_{i,t} = S_{i,t-1} + \tau_S(\delta_{i,t-1} - S_{i,t-1}). \tag{71}$$

Here, $\delta_{i,t} = 1$ when an agent chooses option $i$ at time $t$; $\delta_{i,t} = 0$ otherwise. The outcome $R_t$ is equal to 1 at a time $t$ when a reward is observed and is otherwise 0.

## The DT model outperforms other RL models in model selection and predictive accuracy

To evaluate a variety of different RL models against the DT model, we performed a model comparison and evaluated the comparative predictive accuracies of the various models. Here, we briefly explain how the DT model compared to the other known RL models when computing choice probabilities. These models differ in how they update the value of unchosen options and whether choice history is incorporated into the value update process. The indirect actor model updates the values $Q_{it}$ for each option $i$ in trial $t$ for the chosen options only while freezing the values for the unchosen options. Direct actor model is very similar to average reward rate R-learning models [19] since it updates option values taking into account average reward rates [21]. The direct actor model is also equivalent to the actor critic model if state-space consists of only one state [20]. The forgetting Q *(F-Q)* model, along with updating the chosen option value, also updates the unchosen option value. Thus far, none of these models incorporates choice history as an independent variable. The *F-Q down* model is a version of the F-Q model that discounts the value of the unchosen option with a unique learning rate. The F-Q down model was tested with one learning rate. With two learning rates, the F-Q model is equivalent to the linear–nonlinear Poisson model developed by Corrado and colleagues [24] and shown by Katahira [18]. We tested the alternative possibility that the F-Q model upcounts (the opposite of discount) the action value of the unchosen option (i.e., the *F-Q up* model). Finally, in addition to reward history, the F-Q with a choice trace *(F-Q W/C)* model also incorporates choice history in the selection of upcoming choices. The F-Q model, with and without

a single choice trace, is equivalent to the DT model when $\vartheta = 0$ and $\vartheta = \varphi = 0$, respectively (Materials and methods, RL models for more detailed descriptions).

The model selection criteria were determined based on a test involving the negative log-likelihood. Briefly, the behavioral data (choice and reward histories) were concatenated across all sessions and all subjects and were split into five parts. The model parameters were computed on 4/5 of the data with a 5-fold cross-validation method. The different models were tested on the remaining 1/5 of the data by computing the negative log-likelihood of the choices (see Materials and methods section on the optimization of parameters for behavioral prediction). The prediction accuracy of the models were measured using area under reciever operating curve (AUC) as a metric. This metric, originally developed within signal detection theory, is often used as a binary classification in psychometric tasks with two alternative outcomes [25–27]. The DT model showed a close match to subject's choice dynamics in a representative session (Fig 2D). When various RL models were tested against the mice and human behavioral data (all sessions concatenated), the DT model outperformed all other models (Fig 2E, right panel) in model selection criteria. It also showed best predictive accuracy for mice and second best predictive accuracy for human data (Fig 2E, left panel). The same model comparison tests were also done for the DT model generated data. For this, we first, concatenated all of the behavioral sessions from seven mice (as in Fig 2C), and extracted DT model parameters. Second, we used these parameters to generate the choice and reward history, with the pair of set reward probabilities 0.4:0.1 and 0.1:0.4 for the left and right options, respectively. We also demonstrate that the DT model was effective at recovering randomly selected DT model parameter values from data generated synthetically by the same DT model. For this, we first generated the data using the randomly selected parameters of the DT model and later, using the same model, extracted parameters from the observed data. Parameter recovery was better for learning rates of reward and choice history ($\alpha$, $\tau_F$ and $\tau_S$) compared to temperature ($\beta$) and scaling factors of fast ($\varphi$) and slow ($\theta$) choice traces (S4(A) and S4(B) Fig).

Cross-validation tests for model comparison may favor more complex models [28], which may have biased our model selection in favor of the DT model. To avoid this problem, we also performed variational Bayesian model selection [29] using the MATLAB toolbox to compute protected exceedance probabilities (https://www.github.com/MBB-team/VBA-toolbox). This analysis revealed that the DT model was the best in all model comparison tests. The protected exceedance probabilities for the DT model were 0.6820 and 0.6834 when testing against human and mouse generated data, respectively (S5 Fig).

## Growth of reward probabilities on unchosen options drives choice history effects

We showed that reward history and choice history exert separable effects on current choices. First, we found that first regression coefficients for rewards (mean(SD) = 1.34(0.42)) and choices (mean(SD) = -2.4(1.29)) were significantly different from each other (p = 0.0006, Mann-Whitney U test, n = 7 mice) for the mice and human data (rewards: mean(SD) = 1.39 (1.37); choices: mean(SD) = -1.17(1.93), p = 0.00001, Mann-Whitney U test, n = 19 human subjects). From this finding, it was not clear whether these effects reflected the same or different computational processes. To interrogate what drives choice and reward history effects, we manipulated the reward outcome statistics over three different dimensions, and performed new behavioral manipulations on the new set of mice (n = 4). Specifically, we manipulated one of three behavioral contingencies: experienced total-reward rate, difference in set reward probabilities, and volatility, while holding the other two contingencies constant. Here we define volatility as the length of the block (number of trials) that maintained the same set reward

probabilities. Thus, volatility was inversely proportional to the block length. In addition to reward and choice history effects, we also included reaction time (RT) as a metric of an animal's performance. We defined RT as the time between the animal leaving the center port to the time the animal poked one of the side ports. It has been reported that RTs can be driven by reward expectations [30, 31], while reward history effects are sensitive to the volatility of reward outcomes [32]. However, very little is known regarding what drives choice history effects. Here, we hypothesized that choice history effects reflect the growth of the reward probabilities of the unchosen options. The faster the reward probability of the unchosen option outgrows the reward probability of the chosen option, the stronger the alternation bias should be, and thus, the first regression coefficient of the past choices should be negative. This dynamic can be controlled by manipulating the difference in set reward probabilities.

The effects of volatility on behavioral performance were analyzed by pooling the set of behavioral sessions in which block size was manipulated (4 animals, 44 sessions) and dividing the sets into three groups (terciles) of block lengths of 14–54, 55–96, or 97–136 trials per block in a session. Our results showed that the volatility of the reward outcomes affected the reward history effects. The strongest effects were observed on the first regression coefficient for past rewards (1st tercile mean(SD) = 1.86(1.59), 2nd tercile mean(SD) = 1.24(0.89), 3rd tercile mean(SD) = 0.34(0.38), Pearson correlation coefficient (r) = -0.35, p = 0.0032, 95% CI [-0.54–0.12]), while the regression coefficients for past immediate choices (1st tercile mean(SD) = -3.66(1.9), 2nd tercile mean(SD) = -2.49(1.88), 3rd tercile mean(SD) = -2.95(1.47), r = 0.24, p = 0.043, 95% CI [0.01 0.45]) and RTs (1st tercile mean(SD) = 0.19(0.04), 2nd tercile mean(SD) = 0.2(0.05) 3rd tercile mean(SD) = 0.24(0.04), r = 0.29, p = 0.016, 95% CI [0.06 0.49]) were less affected (Fig 3A and 3B). Different from mice we did not observe any dependency between environmental volatility and the behavior of the Oracle agent (S6(A) Fig). As mentioned previously, this is due to the fact that the Oracle agent is not required to integrate its reward history into future choices because it already has access to the objective reward probabilities of each option.

The effects of the difference in the set reward probabilities (0.8:0.2, 0.7:0.3, 0.6:0.4) on mice performance were analyzed by running animals on a different set of sessions (mice, n = 4; sessions, n = 48). Once more, we partitioned the behavioral sessions into terciles based on the difference in the set reward probabilities. The choice history effects that were analyzed for the first regression coefficients of the mice showed significant changes in response to the difference in the set reward probabilities (1st tercile mean(SD) = -3.66(1.48), 2nd tercile mean(SD) = -3.64(1.78), 3rd tercile mean(SD) = -1.4(2.56), r = 0.42, p = 0.0032, 95% CI [0.15 0.62]), while reward history effects only analyzed for the first regression coefficients (r = -0.15, p = 0.32, 95% CI [-0.41 0.15]) and RTs (r = -0.064 and p = 0.66, 95% CI[-0.34 0.22]) did not show significant changes (Fig 3C and 3D). These effects were also the same for the Oracle agent (S6(B) Fig).

Next, we partitioned the same set of sessions (animals, n = 4 animals; sessions, n = 48) used in the previous analysis into terciles based on the reward rate experienced by the animal per trial. We found that RTs changed as a function of the experienced reward rates, showing a significant and strong positive correlation (1st tercile mean(SD) = 0.14(0.01), 2nd tercile mean(SD) = 0.17(0.03), 3rd tercile mean(SD) = 0.2(0.03), r = 0.51, p = 0.0002, 95% CI [0.26 0.69]) to the overall experienced reward rates (Fig 3F, right panel). The effects of reward (r = -0.15, p = 0.3, 95% CI [-0.42 0.14]) and choice history (r = -0.2, p = 0.18, 95% CI [-0.45 0.09]) were not significantly affected (Fig 3E). These observed effects were counterintuitive because most other studies on this topic have shown a negative correlation between RTs and experienced reward rates [30, 31]. To understand this discrepancy between our results and published work, we must emphasize that the combined (left and right options) set reward probabilities per session were

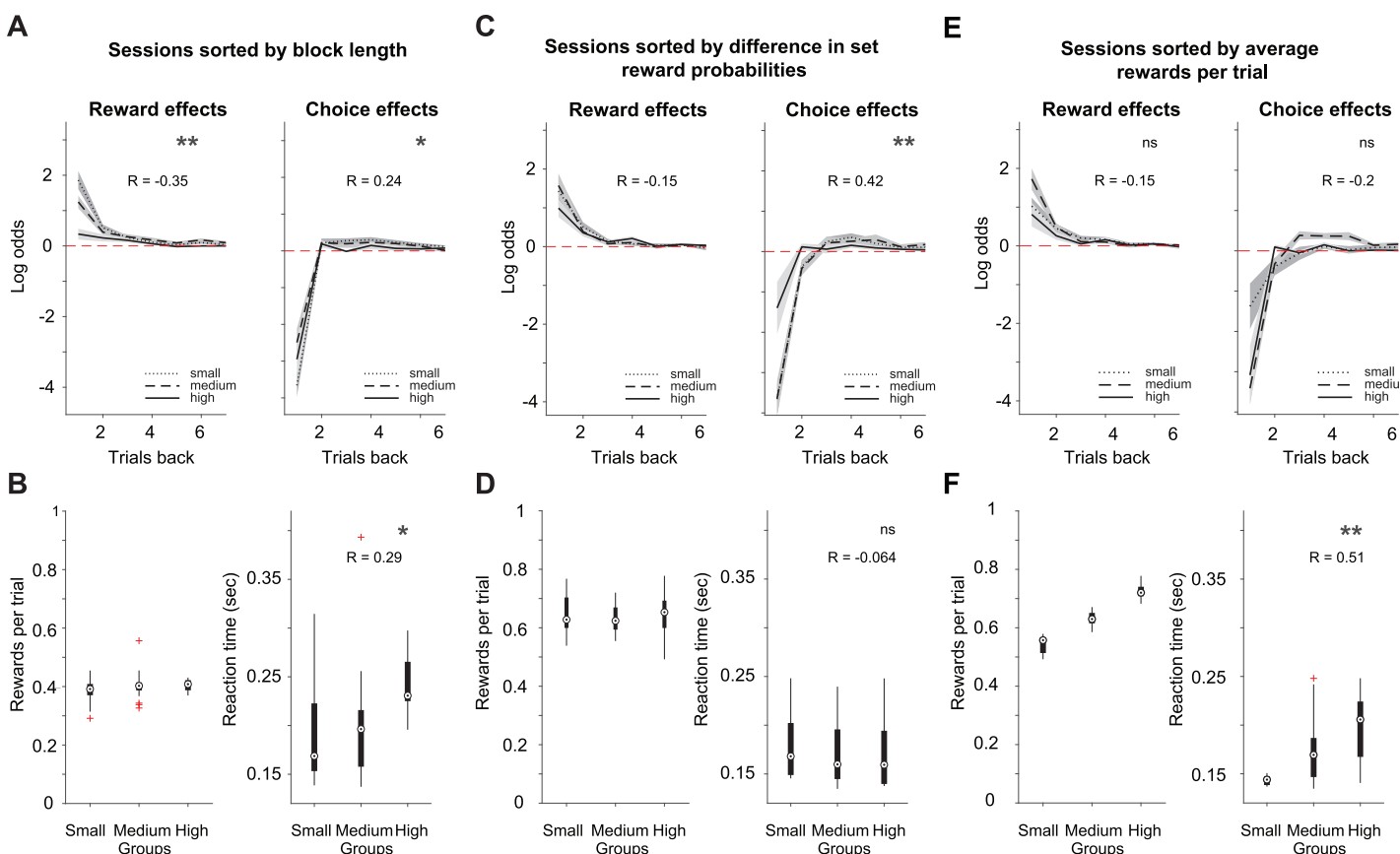

**Fig 3. Volatility, differences in set reward probabilities, and the influence of reward rate per trial on reward and choice history effects.** Sessions sorted into three groups, or terciles (14–54, 55–96 and 97–136 trials for each group), based on the number of trials per block condition (4 mice in 44 sessions). The set reward probabilities were 0.1:0.4 and 0.4: 0.1 for left vs. right options respectively. (A) We show reward (left panel) and choice history effects (right panel) for trials sorted based on block length. (B) shows the average number of rewards per trial obtained by the mice (left panel) and reaction time (right panel). The MATLAB function boxplot depicts median (circle), 25th to 75th percentile of data as edges, all extreme data points with whiskers, and outliers with red crosses. Sessions sorted into three groups based on the difference (Δ) in set reward probabilities (4 mice in 48 sessions). The set reward probabilities pair per session were 0.4:0.6, 0.3:0.7, and 0.2:0.8 for left vs. right and right vs. left options. (C) shows reward (left panel) and choice history effects (right panel) for each group. (D) Shows the average number of rewards per trial obtained by the mice (left panel) and reaction time (right panel) as in B. Sessions sorted in three groups based on the average number of rewards collected per trial (4 mice in 48 sessions). The set reward probabilities pair per session were 0.4 vs. 0.6, 0.3 vs. 0.7, and 0.2 vs. 0.8. (E) shows reward (left panel) and choice history effects (right panel). (F) shows the average number of rewards per trial obtained by the mice (left panel) and reaction time (right panel), as in B and D. The first regression coefficients were used for statistical comparisons across different conditions. The Pearson correlation coefficient of the block lengths, difference in set reward probabilities, and rewards per trial with the regression coefficients of reward and choices one trial back is reported as R within the plots, with their corresponding significance labelled as $p < 0.05 = *$, $p < 0.01 = **$, $p < 0.001 = ***$, and n.s. for a non-significant result.

the same across conditions (for probabilities 0.2, 0.3, 0.4, 0.6, 0.7, and 0.8). Therefore, an increase in the number of rewards per trial experienced by the animals can only be due to the animal's choice strategy. The fundamental structure of the VI task allows the reward rates (rewards per trial) to depend on both the animal's choices and the reward delivery probabilities. We independently validated the negative correlations between the RTs and experienced reward rates by manipulating the overall reward rates and set reward probabilities, as seen in S7 Fig.

Finally, we note that the choice history effects analyzed here (Fig 3A, 3C and 3E) were slightly different from our previous analysis (Fig 1C). Namely, choice history effects for further trials in history tend to zero when we compute the regression coefficients separately for each session and plot their mean, while with the same analysis performed on all concatenated

sessions, we observed that the regression coefficients did not decrease to zero after several trials (Fig 1C). We think that this discrepancy stems from the fact that the number of trials in a concatenated session is much higher than in a single session. Thus, the regression analysis in a concatenated session can reveal more stable effects that last longer among many blocks, even across different sessions.

## The DT model achieves near optimal reward harvesting efficiency using animal-characteristic choice history effects

The DT model is built based on the animal's choice strategy (Fig 2C) and different from optimal models it does not have full access to how the probabilities are updated on unchosen options. Therefore, it was an open question whether an animal-derived decision rule was well-suited to maximizing reward rates in VI type habitats. To test the harvesting efficiency of each RL model, we computed regret as a metric. We defined regret as the difference between the total rewards $R_t$ collected by an agent in a session of $T$ trials and the maximum expected rewards $\mu^*$ [33], Here an agent takes action $a$ during in each trial $t$. Thus, regret can be thought of as a simple cost function for evaluating the optimality of a given strategy.

$$Regret = \mu^* - \sum_{t=1}^{T} R_t = \sum_{t=1}^{T} \max_{a \in A} (P(R|a, t)) - \sum_{t=1}^{T} R_t. \tag{72}$$

The optimization of the DT model was slightly different than that of the other RL models. The animal-derived parameters for $\vartheta$ and $\varphi$ were always constrained to positive and negative values, respectively. However, during the parameter optimization process (that aims to reduce the regret), these parameters can change their sign. To avoid this, we restricted $\varphi$ to take negative values and $\vartheta$ to take positive values, as observed in animals. For all of our analyses we thus used the constrained DT model.

We examined the reward and choice trace of the optimized DT model to see if it retained the characteristic shape of reward and choice history effects observed in animals (Fig 2C). We also tested if these traces reflected the volatility and difference in set reward probabilities as seen in mice (Fig 3). We used the same set reward probabilities as in the mouse experiments in a simulated environment.

First, we manipulated the difference in the set reward probabilities as we did for the mice (Fig 3C and 3D), while maintaining the same overall reward probabilities. Our results show that, generally, the RL models incorporating choice history effects, FQ W/C, and DT models performed better and nearly as good as the optimal RL model (LK model, Fig 4A). A possible explanation of why the LK model does not always outperform any other model is that the LK model needs to estimate the baiting rate because it is built to have more flexible behavior. The latter means that the LK model is able to solve tasks with no baiting (e.g., an armed bandit task), baiting (e.g., the VI task) and tasks that are in between these two conditions. We also noticed that the choice trace of the optimized DT model increased the short-term alternation component as the difference in set reward probabilities decreased (0.2 vs. 0.3) (Fig 4B, right panel). In addition, the shape of the choice trace resembled the choice history effects reported in mice and humans (Fig 2C). Contrary to this we did not observe consistent changes in reward trace as a function of difference in set reward probabilities (Fig 4B, left panel). Based on these results, we conclude that, similar to what we observed in mice, the optimized DT model choice trace reflects the growth rates of the rewards in unchosen options.

Second, we tested the optimized DT model parameters under different volatilities. Again, the RL model with choice trace achieved a near optimal performance (Fig 4C). The slope of

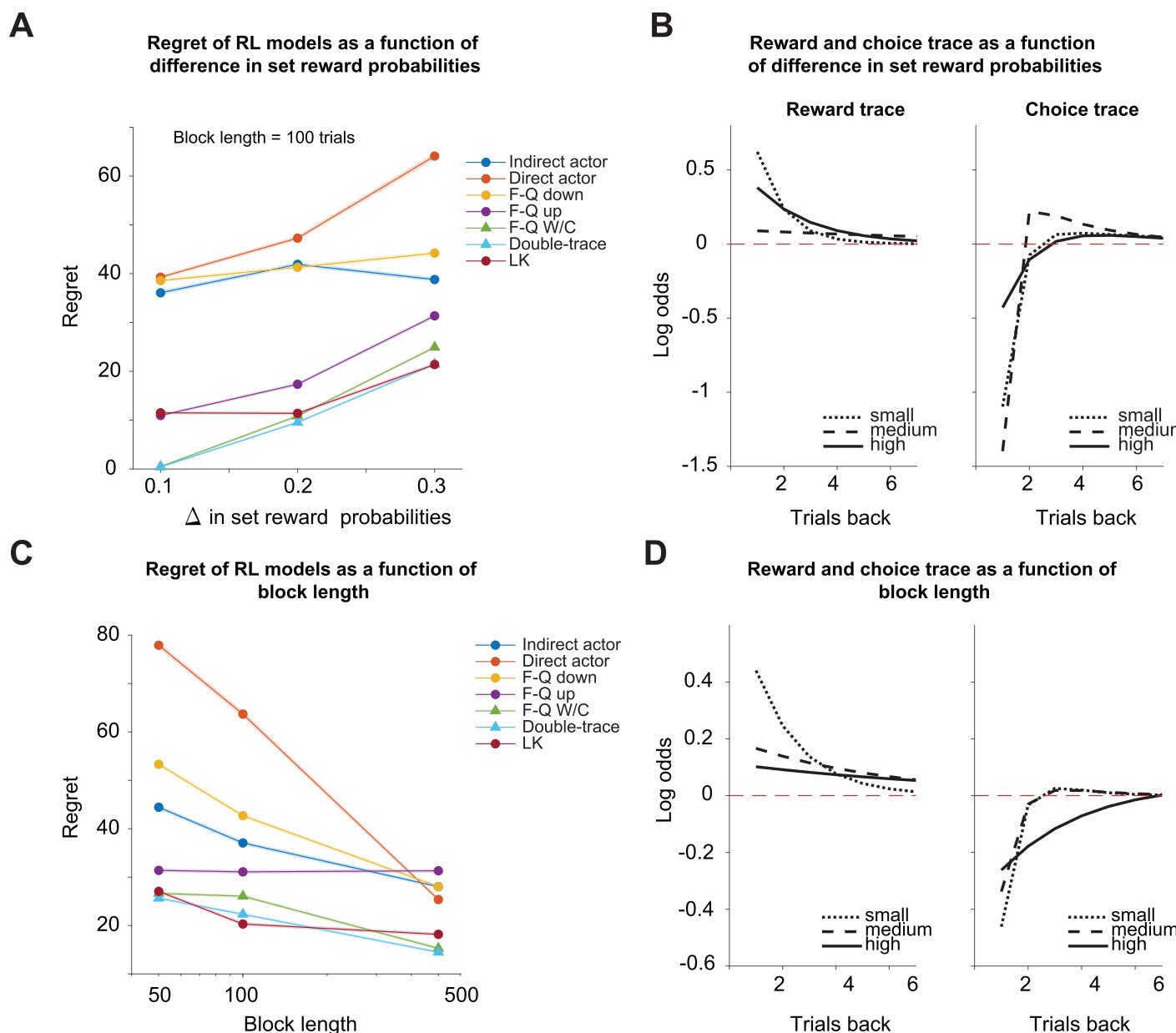

**Fig 4. Harvesting of rewards for RL models, and choice and reward history effects of the DT model.** (A) Regret for different RL models as a function of the difference between the set reward probabilities. The set reward probabilities of the task were 0.45:0.05, 0.4:0.1, and 0.35:0.15 for left vs. right and right vs left options. The block length was fixed to 100 trials. (B) We show the reward and choice traces of the optimized DT model for these pairs of set reward probabilities. (C) Regret for different RL virtual agents as a function of the volatility (block lengths of 50, 100, and 500). The set reward probabilities of the task were 0.1:0.4 and 0.4:0.1 for left and right options respectively. (D) The reward and choice traces of the optimized DT model under these three volatility conditions. For (A) and (C), the regret is defined as the number of rewards away from the optimal collection by the ideal observer in a virtual session with 1,000 trials as defined by Eq (72). The mean is indicated with a solid line, and the standard error of the mean (s.e.m.) is shaded. Note that the s.e.m. is too small to be seen in many cases.

the optimized DT models' reward trace showed a decrement as the block length (inverse of the volatility) increased (Fig 4D, left panel). Different from mice, the choice trace of the DT model also reflected the volatility (Fig 4D, right panel). These inconsistencies could be explained by the noisy recovery of the DT model's scaling factors ($\varphi$ and $\theta$) for the choice trace (S4 Fig).

## Discussion

In this study, we addressed the contingencies between behavior and reward availability, and how these may configure choice history effects. We examined this question from the narrow perspective of biological and synthetic agents that attempt to maximize their reward rates. We showed that similar choice history effects emerge both in synthetic and biological agents, suggesting its adaptive function. This is further supported by our findings and published data in mice, rats, monkeys and humans [2, 13, 17, 34], showing that choice history effects are phylogenetically conserved. These findings lend credence to the hypothesis of a shared evolutionary-selected mechanism that generate choice history effects. Despite their widespread occurrence in diverse species, biological function and behavioral contingencies that drive choice history effects remained elusive. By discovering the adaptive function of the choice history effects, we incorporated it into a new RL variant (DT model) that, when compared to existing models, had a higher predictive adequacy for predicting biological behavior. In synthetic agents the same model yielded higher reward rates, and thus demonstrated a higher energy harvesting efficiency than other candidate models simulated. To further scrutinize the behavioral function of choice history effects, we showed that it tracks the changes in reward probabilities for unchosen options. This effect is distinct from reward history effects, which are predominantly associated with the volatility of the reward probabilities. Choice history effects are also distinct from the reaction times, which [32, 35, 36] reflects the overall reward rates averaged over the options. The choice history effects, as name suggest has been viewed as the outcome of only past choices without any involvement of reward history. However, our findings opens the possibility that choice history effects can be seen as combination of the effects of immediate no rewards and past choices on current choices (S4 Fig). The behavioral response captured by the short-term alternation choice strategy is prevalent in VI tasks without a changeover delay, which punishes the short-term alternation [13, 24]. This suggests that animals use short-term alternation as the default strategy, even though it is optimal only when the set reward probabilities for the two options are very similar. Thus, the short-term alternation choice strategy needs to be explained in our task and others [13, 34] when set reward probabilities markedly differ for two options. Our data suggest that alternation on a short-time scale and perseverance on a longer time scale in choice strategy exist because animals assume a uniform distribution of set reward probabilities. The same choice strategy emerges in the Oracle agent only when set reward probabilities are drawn from uniform distribution. Therefore, animals choices in VI task can be viewed as the outcome of the statistical inference process that future studies can address. The complexity of choice history effects, short term alternation and long term perseverance is captured in the DT model by the fast and slow choice traces. These two different time scaled processes may have different underlying neurobiological mechanisms that we discuss further.

Here, we note a number of limitations that accompany our studies and offer counterarguments that may mitigate these concerns. Although arguably more ecologically valid than experiments in which choice history is decoupled from reward availability, our task could still be limited in its ecological validity. This issue could be remedied by employing more naturalistic foraging fields and by affording greater degrees of motoric freedom when foraging. A number of studies in rodents and humans have explored more ecologically realistic scenarios with innovative task designs [11, 37, 38]. However, the benefit of the more artificial experimental setup of the VI task is the precise control of reward probabilities, while also controlling energetic costs and motoric uncertainties. Estimating the cost of travel or time in foraging animals is difficult, which makes it challenging to compare the performance of animals with that of the optimal agent models. Here, we chose to focus on a simple and straightforward behavioral task

design and used the model derivation to generalize to ecologically relevant contexts. One further concern that could be raised is whether the VI task is really a foraging task. The term "foraging" has been used by some authors to describe decisions between an explicit foreground option against background reward rate of the habitat [39–41]. According to this definition of foraging tasks, an agent typically estimates the average reward rate of the habitat using the R-learning model [10, 19, 37] or follows rules imposed by the marginal value theorem (MVT). Whilst it is true that major theories of foraging posit this as a central problem the animal is attempting to solve, it does not mean that other tasks cannot be defined as foraging tasks. The more broad definition of foraging is simply searching for food sources and making decisions among available alternative options [41–44]. Under this definition, we think that the VI task fits as a foraging task.

## The behavioral function of reward and choice history effects

Animal behavior must be tuned to environmental statistics to maintain its adaptive function [32, 35, 36]. Different dynamics can be discerned in natural habitats that are expected to shape foraging decisions. For example, one such dynamic is the forager experiencing volatility in its food reserves or in the availability of food in its habitat. Another is that there is either positive or negative growth of food availability at unvisited patches. Both of these processes should be incorporated into the decision process to generate adaptive behavior. Previous work [2, 24, 34, 45] that adequately described animals choice and neural dynamics using regression-based and RL-based models have incorporated experienced reward statistics or reward history into the decision process. These models are well-suited for dealing with volatility [32] when unchosen options remain static. However, in VI type habitats, these models [18] (F-Q is one of those models) can be substantially improved when choice history is also incorporated into the decision process. Thus, we argue that reward and choice history effects reflect complementary decision processes tuned to capture the broader reward dynamics in natural habitats.

If choice history effects emerge as an evolutionary adaptation to natural habitats, then the full suppression of this behavior, even in well-trained animals, in a perceptual or value-based decision-making tasks might be difficult. Indeed, in many decision-making tasks that do not impose contingencies of previous choices on current choices, choice history effects persist [1, 3, 6, 22, 46]. Understanding the behavioral mechanisms that generate choice-history effects in these tasks may require the manipulation of behavioral contingencies, linking past choices to current choices in a parametric way. Even if behavioral mechanisms that generate choice history effects are hidden, models that incorporate these effects in the decision process may be more adequate in describing behavior. Lapses in perceptual decision-making tasks, are commonly observed in which animals do not select the most rewarding action. These lapses could be explained by animals assuming that their current environment is dynamic, and thus making it valuable to explore less favorable options from time to time [47]. It would be interesting to apply a modified version of the DT model to perceptual decision-making tasks to attempt to capture such lapse behavior. Furthermore, in perceptual decision-making tasks that expose animals to a static environment, models that assume a dynamic environment are better at capturing the animals' choices [48]. However, when an environment is not static and task-relevant stimuli exhibit autocorrelations or bias, choice history effects carry an important adaptive function [49–51]. Different from perceptual decision making tasks explicit manipulation of behavioral contingencies that would reveal the function of choice history effects in reward foraging tasks to our knowledge have not been reported. However, theoretical work [52] shows that for patch foraging agents the optimal strategy should take into account the replenishing rates of unvisited patches. Supported by experimental work [53] manipulation of replenishing

rates on different unvisited patches results in more frequent visits to fast replenishing sites as well as early leaving times in human subjects. While these studies did not examine the role of the choice history effects in maximizing the reward harvesting efficiency, these findings are consistent with our data that behavior of animal (choice history effects) is sensitive to growth rate of probabilities on unchosen options. The experimental data showing that neural representations of choice history are widely distributed in sensory, motor, and association areas [4, 54, 55], further suggest that these representations carry adaptive function.

## Implications for neural representations

In terms of choice history effects, the possibility that short-term alternation is driven by no rewards, and that long-term is driven by previous choices, raises questions regarding their neural representation. Are these slow and fast processes encoded by distinct neural populations? Or do they emerge from the same neural circuits at different time scales by virtue of mixed selectivity [56]? The findings that no reward responses or short-term alternation of choices in prefrontal cortical areas drive behavioral adaptation to task contingencies supports the hypothesis that dedicated neural circuits encode the faster, no reward component of choice history effects [57–60]. We are not aware of any published work that describes the single neuron correlates of the choice history for past two to 15 trials captured by the slow component of choice history effects. However, slow and fast choice history single neuron representations, albeit at the different time scales (slow history effect comprised trials past within the 130–400 range), have been described in monkey prefrontal areas [46]. This suggests that the same neural circuits can capture both time scale processes in VI task. The two time scales of decision processes appear in many reward foraging and perceptual tasks. In sequential decision-making tasks, normative models based on Bayesian principles [61] reveal within- and across-trial accumulation processes that operate on two different time scales [62]. The two time scales of value accumulation also appear in optimal models with foraging tasks that provide fixed (i.e., non-replenishing) reward probabilities [63]. Furthermore, imaging studies have revealed multiple time scales of learning in humans [22, 64–67]. Thus, fundamental questions emerge as to how individual neurons capture the multiplexed nature of decision-making phenomena, short-term alternation (or no reward responses) and long-term perseverance in VI task.

## DT model and sequential foraging tasks

What benefits could choice history effects and its implementation in the form of an RL model provide to foraging agents that make decisions between engaging with the current option or searching for better alternatives? The two major classes of models can formalize these type of decision-making processes. Models based on the MVT state that optimal agents should leave the food patch when the reward rate of that patch drops below the average reward rate of the habitat [41, 68–70]. Thus, MVT assumes that the agent knows the average reward rate over the entire habitat. In contrast, agents can, in principle, learn the average rate of rewards of the habitat by exploration and experience. Indeed, a classes of models that estimate the average reward rate and use that estimation to update policy succeed at explaining both the subject's behavior and neural dynamics in frontal cortical areas [66, 67, 71, 72]. In the VI task, however the same class of model (direct actor RL) failed to predict subjects choices and was inferior in its reward harvesting efficiency compared to the DT model. Furthermore, using a model free decision theoretic approach, when we explicitly manipulated average reward rates, we failed to see any effect on choice or reward history. This was also supported by the regression analysis that failed to reveal any effect of global reward rates on choices. Thus, based on our data, we suggest that in the the VI tasks, agents use different

heuristics than those afforded by R-learning models or models inspired by MVT. It would be interesting to compare DT model with other R-learning models in terms of reward harvesting efficiency in sequential foraging tasks.

## Conclusion

Here we show that mice and humans can express similar choice history effects when searching for rewards in habitats where past choices effect future reward availability. By simulating agents in equivalent environments, we found that the same choice history effects yielded gains in the efficiency by which rewards are harvested. The double trace model that we propose, which incorporates choice history explicitly in its architecture, was advantageous over competing models, both in terms of its performance in predicting choice behavior, and in terms of the efficiency with which it can harvest rewards. All elements combined, we provided an initial explanatory and algorithmic account of choice history effects beyond their description as a bias, connecting this concept to a broader class of optimality models within behavioral ecology.

## Supporting information

**S1 Fig. Fraction of choices in the VI task and performance of virtual agents.** (A) Fractional difference between consecutive choices and alternation on the last alternation trial in a two-alternative task under the baiting schedule (VI task) where the set reward probabilities of each option are taken randomly between 0 and 1 from a uniform distribution. Fractional difference was computed by subtracting the consecutive choices from alternation and dividing that difference on the total number of choices on each trial from the entire simulation set (n = 200000). (B) Average regret (missed rewards collected with respect to the Oracle agent reward collection) per trial in 100 sessions of an agent making random choices, the IB agent and the LK model. Sessions had $T \in [450, 1350]$ trials, corresponding to 9 blocks with block size $\in [50, 150]$ trials selected randomly from a uniform distribution. Significance labeled as $p < 0.05 = ^*$, $p < 0.01 = ^{**}$ and $p < 0.001 = ^{***}$ using non parametric Mann-Whitney test. (C) Recovery of set reward probabilities by the IB agent. The IB agent was tested using set reward probabilities of 0.9 vs 0.1, 0.7 vs 0.3 and 0.6 vs 0.4. For each pair of probabilities we used 30 sessions that consisted of 1000 trials. Black dots indicate inferred set reward probabilities for 30 sessions.
(EPS)

**S2 Fig. Reward and choice effects of individual subjects.** (A) Reward and choice history effects analyzed by LASSO regression analysis for 19 human subjects. Each line indicates an individual subject. (B) The reward and choice history effects shown for individual mice (n = 7).
(EPS)

**S3 Fig. Reward and choice history interaction.** (A) LASSO regularized regression analysis (30 trials back) was carried out on past rewards unlinked to the rewarded option (denoted as unsigned rewards here), past choices, past right rewards, past left rewards, past right no rewards and past left no rewards in that order. Six panels show concatenated sessions from 7 mice as in Fig 2C. (B) The same analysis was done for individual animals. Mean is shown with solid lines and and s.e.m. is shown with shaded blue color.
(EPS)

**S4 Fig. Parameter recovery for the DT model.** (A) The DT model with one set of randomly selected parameter values (shown with blue horizontal bars) was run 200 times in a session of

1000 trials. Each session consisted of a block of trials with the length of 100 trials. Each block had a pair of set reward probabilities 0.1:0.4 and 0.4:0.1 for left and right options respectively. Next to the true parameter values (blue bar) boxplots (median—red bar, edges 25-75 percentile and whiskers show all data points except outliers) and all 200 parameter values (black dots) recovered by the DT model. (B) The same parameter recovery was done for 1000 randomly selected parameter values, except that this time we performed recovery only once. Scatter plot shows on x axis the initial parameter values and y-axis is the recovered value. All insets in lower 3 panels show zoomed version of the parameter values restricted to the full range of true parameter values for x and y axis. The initial parameter values were restricted to the following range: $\alpha \epsilon [0\ 1]$, $\tau_F \epsilon [0\ 1]$, $\tau_S \epsilon [0\ 1]$, $\beta \epsilon [0\ 4]$, $\vartheta \epsilon [0\ 4]$, and $\varphi \epsilon [-4\ 0]$. r indicates Pearson correlation coefficient. All p values were below 0.001.
(EPS)

**S5 Fig. Bayesian model selection.** Numbers on x axis indicate different models. 1—Indirect model, 2—F-Q, 3—F-Q up, 4—F-Q W/C, 5—Double trace, 6—LK model, 7—Generalized linear model (A). Humans, (B) mice.
(EPS)

**S6 Fig. Difference in set reward probabilities affects the choice history effects of the Oracle agent.** (A) Sessions of the Oracle agent sorted in three groups by the block length. We used blocks of 50, 100 and 500 trials for 150 sessions with 1000 trials for each session. The set reward probabilities pair per session were 0.1:0.4 and 0.4:0.1 for left vs. right options. We show their reward and choice history effects. (B) Sessions of the Oracle agent sorted in three groups by the difference (Δ) in set reward probabilities (150 sessions with 1000 trials each). The set reward probabilities pair per session were 0.4:0.6, 0.3:0.7 and 0.2:0.8 for left vs. right options and right vs. left options. We show the reward and choice history effects. The correlations of the first regression coefficients as a function of block length or difference in set reward probability for reward and choices are reported as R (Pearson correlation coefficient) with their correspondent significance labelled as $p < 0.05 = ^*$, $p < 0.01 = ^{**}$, $p < 0.001 = ^{***}$. Here n.s. states for a non-significant result.
(EPS)

**S7 Fig. Increase in the total probability of reward delivery decreases the reaction time of mice.** Sessions of four mice sorted in three groups (terciles) by the mean in set reward probabilities (62 sessions), or in other words, the mean set probability of rewards in a session. The set reward probabilities of the leaner side was kept to 0.1, while the richer side was set to 0.4, 0.5, 0.6 or 0.8 in a session. (A) Shows their correspondent reward history effects, choice history effects. (B) The average number of rewards collected per trial (left panel) and reaction times (right panel). The correlations of the block lengths with the coefficients of reward and choices one trial back are reported as Pearson correlation coefficient (R) on the plots with their correspondent significance labelled as $p < 0.05 = ^*$, $p < 0.01 = ^{**}$ and $p < 0.001 = ^{***}$.
(EPS)

**S1 Video. Human task.**
(MP4)

**S2 Video. Mouse task.**
(MP4)

**S1 Appendix. Description of all the notations and symbols.**
(TIF)

## Acknowledgments

We thank Larry F. Abbott and Ashok L. Kumar for their suggestions on the DT model and the manuscript. We thank Sophie Seidenbecher and Madeny Belkhiri for their assistance with editing the manuscript. We thank Søren Rud Keiding for his advice and discussions, Eske Nielsen for programming the human game in Unity platform and Maris Sala and Daniel Kozlovski for assisting with the data collection.

## Author Contributions

**Conceptualization:** Junior Samuel López-Yépez, Duda Kvitsiani.

**Data curation:** Junior Samuel López-Yépez, Juliane Martin, Duda Kvitsiani.

**Formal analysis:** Junior Samuel López-Yépez, Duda Kvitsiani.

**Funding acquisition:** Duda Kvitsiani.

**Investigation:** Juliane Martin, Duda Kvitsiani.

**Methodology:** Junior Samuel López-Yépez, Juliane Martin, Duda Kvitsiani.

**Resources:** Junior Samuel López-Yépez, Juliane Martin, Duda Kvitsiani.

**Software:** Junior Samuel López-Yépez, Duda Kvitsiani.

**Supervision:** Duda Kvitsiani.

**Validation:** Junior Samuel López-Yépez.

**Visualization:** Junior Samuel López-Yépez, Duda Kvitsiani.

**Writing – original draft:** Junior Samuel López-Yépez, Juliane Martin, Duda Kvitsiani.

**Writing – review & editing:** Junior Samuel López-Yépez, Juliane Martin, Oliver Hulme, Duda Kvitsiani.

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
