## [Decision Letter · Decision Letter 0]

2 Mar 2021

Dear Dr Kvitsiani, 

Thank you very much for submitting your manuscript "Choice history effects in mice and humans improve reward harvesting efficiency" for consideration at PLOS Computational Biology.

As with all papers reviewed by the journal, your manuscript was reviewed by members of the editorial board and by several independent reviewers. In light of the reviews (below this email), we would like to invite the resubmission of a significantly-revised version that takes into account the reviewers' comments.

As you will see, although encouraging, all reviewers point out several issues that should be fully addressed. I would like to attract your attention on the parameter recoverability issue raised by two reviewers. I also would like to add that a model recovery analysis is warranted in your case.

All reviewers pointed out several instances where the paper crucially lacks clarity or non-standard methods are used. We encourage your to better clarify these issues and justify the methods.

We cannot make any decision about publication until we have seen the revised manuscript and your response to the reviewers' comments. Your revised manuscript is also likely to be sent to reviewers for further evaluation.

Sincerely,

Stefano Palminteri

Associate Editor

PLOS Computational Biology

Daniele Marinazzo

Deputy Editor

PLOS Computational Biology

As you will see, although encouraging, all reviewers point out several issues that should be fully addressed. I would like to attract your attention on the parameter recoverability issue raised by two reviewers. I also would like to add that a model recovery analysis is warranted in your case.

All reviewers pointed out several instances where the paper crucially lacks clarity or non-standard methods are used. We encourage your to better clarify these issues and justify the methods.

Reviewer's Responses to Questions

**Comments to the Authors:**

Reviewer #1: In this study, the authors investigated the choice history effects in humans and mice during choice behavior reinforced with the variable interval (VI) schedule. They claimed that the observed choice history effects have an advantage in foraging behavior in a natural foraging environment. Given that the choice history effects have attracted growing attention, studies on their adaptive functions are of much interest. But I think there are many issues to be fixed in this paper: There are several places where the methods are not clear, and the validity of the models and analysis is somewhat doubtful. Perhaps this is due to the author's use of their customized method without being fully aware of standard modeling methodologies. I think that the authors should make sufficient efforts to fix these issues. Below are the main issues, especially regarding the methods. But if the authors are willing to revise the manuscript, I recommend that the authors review and revise the entire manuscript.

1. Task description

The behavioral task descriptions are insufficient, so the readers cannot obtain information to reproduce the experiments. For example, the paper contains no information about the sex of the subjects (of both mice and humans). How many trials did Mice experienced per session? How long is each session? How did the authors create the program of the computer game for human experiments? Was this done online or in a lab? Why are the trial numbers different across subjects? Also, there is no information on the experimental equipment for mice.

2. Models

line 571- The Oracle agent

- I could not follow why Eq (7) become looks like this. Where did “(1 – P(R|I,t))(1 - \\delta_{I,t})” come from? What exactly does "during block transition" (line 575) mean? If this is an expression that depends on the task design, it should be explained in the task description instead of this section.

line 577- The Bayesian agent

- I'm not sure if the update rule of this model is a calculation that can be called "Bayesian." At first glance, it seems to perform a sampling-based approximation to Bayesian inference, like the sequential Monte Carlo method or the particle filter. However, looking closely, that is not the case. Eq. (12) certainly follows Bayes' formula, but using it for each samples of the parameter does not necessarily lead to that the samples of θ approximate a Bayesian posterior distribution. Also, I cannot figure out how adding a noise factor (line 596) will affect the distribution. If there is research using such a method, the reference is required. If it is the author's original, it is necessary to explain why this update rule should be Bayesian.

line 604- Optimal RL agent (LK model)

- It is unclear to me in what sense this LK model can be called an optimal RL. What does the LK model stand for? I could not figure out why the Qset updated by Eq (16) equals P_set. I could not follow subsequent calculations either.

line 636 “The learning rate should then be affected by the prediction error of the baiting rate.”

- It is unclear why this can be said and what kind of logic will be used in the subsequent update rule, Eq (20).

line 643- Linear regression for computing reward and choice history effects

- It is difficult to understand whether Eq (21) is an equation (How do the left and right sides correspond? Why the parameter \\lambda is included in the left side while not in the right side?). It is also necessary to explain why L1 regularization and L2 regularization are used instead of the standard logistic regression.

Eq (22) is an equation that can be applied to any number of choices, A, but it should be noted that Eq (23) holds when there are only two choices.

lines 670-680

It seems that maximum likelihood estimation is performed with an original method, but why did not the authors use the established and packaged nonlinear optimization method (which can be executed by the function fmincon in Matlab)? Since it seems that the gradient of the objective function has not been estimated, it might take extra time to converge. If there is literature that examines the validity of the method used here, it should be cited.

lines 681-687

Also, I have never seen a document in which AUC is also used in this way. I don't know how to calculate it. I think it needs an explanation. Authors should cite any references available.

line 712- Derivation of the DT model from logistic regression

- The DT model consists of a combination of two choice kernels with fast decay and slow decay. I think this can be introduced quite naturally. I'm not sure why it needs to be derived from logistic regression, as the authors discussed. I think it makes sense to associate it with logistic regression for a deeper understanding of the model property, but I'm not sure if the derivation here works for that purpose.

p.46- RL models

I understand that “Indirect actor” is a natural Q-learning update, but I couldn’t understand how the update rule for the “Direct actor model” (64) was derived. An explanation is necessary.

p.48

“F-Q up model” is equivalent to the RL model with “default value”, discussed in Toyama, Katahira, & Ohira (2019, Frontiers in Human Neuroscience), albeit this paper assumes that the initial value of Q is set to the default value.

Related to this, the authors should mention how the initial value of Q was set.

3. Results

Eq (1) (p.6)

Why is “P_set, i (R)” denoted like a function of R? Isn't this a constant?

line 211 “The discrepancy in choice history effects of optimal agents and animals must stem from the fact that optimal agents can precisely infer the reward probabilities while animals can not.”

- The “discrepancy” mentioned in this sentence is not clear. I think it would be better to show the actual data rather than refer to Lau & Glimcher (2005).

line 233

How the data were split into five parts? Did the authors split sessions within each subject? Or did they ignore the subject identity and pool all the sessions?

line 270

“This finding was also evident when looking at the representative choice dynamics that the DT model, mice, and humans generated (Fig. 2E)."

- I could not figure out how this was evident from Fig.2E.

Table 1 (p.16)

Only a single simulation of parameter recovery was reported. Parameter recovery should be done many times while changing the true parameters. Please see the following paper:

Wilson, R. C., & Collins, A. G. (2019). Ten simple rules for the computational modeling of behavioral data. elife, 8, e49547.

line 380

“Due to the symmetric nature of the two choice traces, whereby different signed parameters can yield the same predictions, we restricted \\varphi to negative and \\vartheta to positive values.”

- I think this statement is inappropriate. Because \\varphi was defined as the weight for *first* decay and \\vartheta was defined for *slow* decay, it is not symmetric.

How did the authors determine model parameters for Figure 4? (Was it selected to minimize the regret?)

4. Other points

Throughout this paper, the authors argue that VIs are more ecologically valid. But I do not think this is always the case. While this may be the case for herbivore foraging, the variable ratio (VR) schedule, in which rewards do not depend on choice history, is more appropriate for carnivore foraging (see Sakai & Fukai, 2008, Neural Computation).

Also, it would be interesting to analyze the data in terms of whether it follows the matching law. (e.g., Does the choice history effect promote or inhibit matching behavior?) As Sakai & Fukai demonstrated that actor-critic model (but not Q-learning model) can explain the matching law, it may be better to include an actor-critic model as a candidate model.

Reviewer #2: General comments:

Junior Samuel Lopez-Yepez and his colleagues are interested in the issues of choice history effects. The authors are interested in the “decoupling” of reward and choices in the sequence of trials. In the natural habitat, the choices act on the environment and change its state, which subsequently influences the reward richness of the environment. But such inter-trial dependency of the choices and rewards are not carefully examined in the previous studies. They devised the task paradigm with a variable interval (VI) reward schedule, which incorporates such inter-trial dependencies of choices and rewards. By examining the behaviors of humans and mice on this task paradigm, the authors have found that application of the double trace (DT) model, which was originally applied to the reward history effects of monkeys, to choice history effects can explain the behavior of animals well. Therefore, the value of the paper depends on how the VI task paradigm is novel and in introducing the intertrial reward-choice dependencies and on how the DT model is adequate in describing and explaining the interactions of choice and reward in long-term scales beyond a single trial. Overall, there is some puzzling aspect in this paper. The conclusion of the paper is just about the choice history effects but the task design and the initial aim of the paper is so explicitly set out to examine the effects of the long-term contingency of choice and rewards. The authors need to address the disconnections in the design, analysis, and the interpretation in this study.

Major points:

1. The first issue is how much of this new task paradigm is effective for elucidating the effects of long-term choice-reward contingency compared to the previous paradigms. As the authors mentioned, the task paradigms inspired by foraging theory has uncovered interesting patterns of the long-term history effects of choice and reward (Hayden et al., Nature Neuroscience, 2011; Kolling et al., Science, 2012; Kolling et al., Neuron, 2014; Wittmann et al., Nature Communications, 2016; Wittmann et al., Nature Communications, 2020). The authors should conduct more careful comparisons of these existing literatures.

2. Related to the first point, the previous literature of foraging decision paradigms with long-term contingency of choice and reward elucidated mainly the history effects of rewards in the previous trials (Wittmann et al., 2016; Wittmann et al., 2020). In contrast, the authors claim that the long-term history effects are mainly due to the choice history effects. It is crucial to reconcile the difference of these interpretations between the current results and the previous results using the conceptually similar tasks.

3. Separation of the choice history effects and reward history effects in the behavioral analyses are basically artificial in the current study and in the previous studies. This issue is more pronounced in this study because authors explicitly aimed to examine the effects of the across-trial contingency between the choices and rewards. The analyses in this study, however, linearly separated history effects of choices and rewards. It might be more straightforward to examine such “interaction terms” of choices and rewards if the authors are genuinely interested in these intricate relationships between choices and rewards.

4. DT model of “choice history” effects might contain the similar problem of the artificial separation of the choice and rewards. The author’s focus on the choice history effects is understandable given the recent popularity of the issue of the choice history effects or decision inertia. But it is also puzzling that the conclusion of the paper is just about the choice history effects given that the task design and the initial aim of the paper is so explicitly set out to examine the effects of the long-term contingency of choice and rewards.

Minor points

1. The article uses concepts or variables (alternation, perseverance, volatility) without introducing them first, which is pretty confusing, even if those variables can be more or less intuitive. The results section did not give enough information for the analyzed variables to be intuitive enough. I needed to go back and forth from the results to the methods constantly, and sometimes to find that the methods did not contain the information I was looking for either.

2. Figures are also incomplete and not everything that appears is labeled or explained. The text gets particularly confusing when dealing with the particular dataset each analysis was based on. Some were based on simulations (optimal agents or their proposed model), while others are based on mouse and human data. In the case of simulated agents, there are several reward distributions tested for optimal agents, and it looks like sometimes block transitions are used and sometimes not. In the case of biological data, two mouse datasets are used, and most of the analyses are not done with the human dataset at all. The authors should make an effort to clarify the particularities of the environment data was simulated/collected on, and to justify why each kind of analysis was done on only certain datasets.

3. In the authors summary (page 3), authors make clear what process their model captures which other models do not. Namely, and in their own words, the fact that “animals track their history of visits and use it to maximize the food harvesting efficiency". To me that makes a lot of sense. However, it is difficult to find this clear message in the main text, particularly in the discussion.

Specific points

PAGE 3

• Lines 55-57: the authors suggest that “choice history bias” is synonymous with “choice inertia”, while that is not exactly true. Inertia is a particular kind of bias where the probability of repeating a choice increases the more a choice is repeated (Akaishi et al., Neuron, 2014).

PAGE 4

• Lines 71-72: I find the way the reward probability in the VI task is explained a bit confusing. Maybe it is because it starts by saying “in each trial rewards are assigned with fixed or set reward probabilities keyed to different options”, and it is not until much later that it is said that the probability also depends on the number of elapsed trials without that option being chosen. Maybe authors should start by saying that each option’s reward probability is defined by both a set (or initial) probability, unique for each option, and the number of trials without that option being chosen.

PAGE 5

• Lines 89-90: The affirmation “To maximize their reward rate, optimal agents should choose the options with the highest set reward probability” assumes that both options lead to a reward of the same magnitude, if this is delivered (e.g. 1 point). Maybe this should be stated before when describing the VI task, as many decision-making studies implement different rewards for each option.

• Lines 92-93: The authors say that an optimal strategy in this environment may “resemble choosing the best option, with occasional switches to lesser options”. This statement is not well connected to the remaining part of the paragraph, but it could make an interesting point to be developed in the discussion. It would be particularly interesting to connect it to what previous work thinks such lapses mean (i.e. https://elifesciences.org/articles/55490).

PAGE 8

• Lines 171-173: The peaks observed in simulated data should be discussed a little bit.

PAGE 16

Lines 303-304: a short definition of volatility should be introduced here, as it is important to understand the following analyses. The authors need to explain what “reward history effects are sensitive to the volatility” means. In the original paper, it was learning rate, which was affected by the volatility (Behrens et al., Nature Neuroscience, 2007).

PAGE 20

• Figure 3: the figure and the legend need serious reworking. Why are the middle panels on the left hand side of A, B and C needed? They simply depend on experimenters’ fixed parameters. What do yellow dots mean? What kind of variability are we being shown in each panel with those error bars? What do red crosses mean? What do blue and grey circles represent on the right hand side panels? Also it should be said that the volatility analyses were made on a different dataset than the rest of the analyses described in the figure, for which new data (although with the same participants) was gathered.

PAGE 22

• Lines 410-411: why do authors think that the DT model’s choice trace reflects volatility, but they do not find that in data?

• Lines 414-436: the first paragraph of the discussion is way too long and it does not summarize the achievements of the present work. It only gives context, and it would be more appropriate for the introduction.

PAGE 23

• Lines 424-432: the authors take a convoluted path to make a simple point, which, if I understand correctly, is the following. Some experimental work whose primary aim is studying optimal behavior, but not necessarily within the domain of perception, still resorts to perceptual tasks. But assessing optimality in perceptual tasks has extra nuisances that we could disregard in simpler, non-perceptual scenarios, such as the one proposed here. I feel like this point could be made easier.

• Line 435: what property are the authors talking about?

PAGE 24

• Lines 442-444: authors say that “optimal agents show the same characteristic shape of choice history effects observed in animals when probed with uniform distribution of set reward probabilities”. I think this is quite confusing. The key information to understand this is to bear in mind that the optimal agents the authors are talking about have access to the exact set probabilities of each option. Thus, in this case their “knowledge” transparently reflects the experimental parameters. Maybe authors want to make that clear.

• Lines 445-447: I do not see how point #2 falls from what the authors have previously reasoned in that paragraph.

PAGE 25

• Lines 480-483: I feel this is an overstatement. Just because these models take reward history into account it does not mean they will offer an accurate description of behavior and neural activity.

• Line 483: why the “however”?

PAGE 26

• Lines 502-515: this is a paragraph about neural representations, and yet no neuroimaging work is cited.

• Line 507: sometimes authors spell “time scales”, sometimes “time-scales”. A unified spelling should be used.

PAGE 27

• Lines 516-534: Charnov’s article aside, this paragraph only has one citation. Authors should try to cite more sources.

• Line 536-540: the conclusion should succinctly summarize the most important points of the manuscript, yet the authors use it to suggest, for the first time, that choice history effects are phylogenetically conserved. Maybe this evolution-related bit should be moved to the discussion, and the conclusion should focus on a short wrap-up.

Reviewer #3: Lopez-Yepez and colleagues examine the role of choice history in decision making using a binary 2AFC task with a Variable Interval (VI) Reward Schedule; this introduces contingency between the time spent since an option was chosen and likelihood of payout if the option is chosen again. This is different to a lot of 2AFC tasks in which choice biases emerge despite their being no built-in dependency between past choices and current prospects. They find that the choices of animals and humans are influenced by both the history of rewards and history of choices. They also find in mice that the effects of reward history are mediated by volatility (defined as length of the block) whilst effects of choice history are mediated by differences in set reward probabilities. A learning model that incorporates the history of choices is shown to be able to recapitulate these effects, confer advantage over various other models and shown to earn rewards that are closest to maximum possible (indexed via regret) suggesting that incorporating this knowledge helps agents maximise returns.

I like the task and applaud the authors attempt to provide a plausible account of choice biases that are often observed in tasks.

My main set of (related) issues are that in quite a few places, some things about the design and the analysis came across as unclear or confusing.

An example is the experiments the authors run – with mice they run 3 conditions (0.25:0.25, 0.40:0.10 and 0.10:0.40) but with humans they just run 2 conditions (0.40:0.10, and 0.10:0.40). Already it is confusing why this difference in designs exists (no explanation is provided from what I could see). But to confuse things further, in the figure (Figure 2C), it suggests that humans actually did also have the 0.25:0.25 condition? I put a few more examples below. Note – these may each be things that can be tidied up easy enough, but combined it gave me the impression of being a bit careless.

A few other examples (not an exhaustive list):

• Optimal Agents: The analysis of choice history for the 3 optimal agent simulations (Fig. 1) is presented in a haphazard way.

e.g., why have the number of trials/sessions be different for each combinations of set reward probabilities for the Oracle Agent (P.7)? [For instance, in (1) Number of trials is 1000 and number of sessions is 10 but in (3) Number of trials is either 450 or 1350]

e.g., why have n=100 sessions for Bayesian agent, n=5000 for the LK agent and n=10 for Oracle?

e.g., why use a greedy rule for the Oracle but a softmax for the Bayesian and LK Agents?

[Maybe there are good reasons for these differences, but it wasn’t obvious to me.]

• Parameter Recovery (Table 1): There are large details missing on how this procedure was carried out. For instance:

how many simulations were run?

why were these specific DT parameter values selected (e.g., are they the average of the parameters fit to the mice data, the human data, both?)

How many trials/blocks were used in the simulations (was it like the human task or the animal task, for instance)

Is it possible to get estimates of variability for the recovered parameters?

• Figure 1B – I think the legend is incorrect as the option with the higher probability (blue line) actually has lower probability of reward for each number of unchosen trials.

Other issues:

• Terminology: The authors describe their approach and task as one that studies “foraging behaviour” (ln97) and explores “foraging decisions” (ln437). Whilst incorporating a nice feature - that time since a specific option was last chosen influences likelihood of its future reward – which is more like some real situations faced by animals outside of the lab, this is definitively not a foraging task. The key premise of foraging tasks (such as patch foraging or prey selection) is to have one explicit foreground option that needs to be considered against an estimate of the background reward rate. This is just not the case here; the task used is a 2AFC binary choice task where agents are given an explicit “menu” of all the options available on each trial. See for instance:

o Hayden, B. Y., & Walton, M. E. (2014). Neuroscience of foraging. Frontiers in neuroscience, 8, 81.

o Hall-McMaster, S., & Luyckx, F. (2019). Revisiting foraging approaches in neuroscience. Cognitive, Affective, & Behavioral Neuroscience, 19(2), 225-230.

o Stephens & Krebs, Foraging Theory

• Task (humans): There are only 19 participants in the human task and 20-30 trials per block. This seems very few trials to be able to fit a learning model reliably. Did the authors conduct any checks for this?

• DT Model. It seemed to me a bit peculiar to include both a fast trace and a slow trace of choice history in the model in so far as these quantities have the same update applied to them on every trial. I understand that they end up being different quantities all the same owing to the learning rates being high and low respectively and potentially this enables their model to capture both the fast trial to trial oscillations in the choice effects (e.g., Fig 2c) as well as slower trends over time. Nonetheless, did the authors do any work to untangle whether you need both these traces in their model? (e.g., is one trace doing most of the work in explaining choice history, could a single trace model beat a double trace model?). One option might be to run Model Recovery – simulate data from 3 models (No trace, One Trace, Two Trace) and see which of the 3 models is best fit by the data in each case. For instance, when simulating choices from a model in which there is no trace history incorporated, this should be the winning model (e.g., determined by Bayesian Model Selection) when you fit the 3 contenders to the choices.

• There seem to be some differences in how well the DT model recovers the history effects. For different set probabilities (figure 4a), for the small set probabilities condition, it seems the DT model has a positive effect after a few trials back – but this is not the case in the data (Figure 3b) where effects seems to be asymptote after a few trials at around 0. The size of the effects also seem to differ by a lot. Compare the scale of the y axis on 4a Choice Traces plot (ranges from -1.5 to 0.5) to that of 3B Choice effects (range -4 to 2). Similar differences emerge when you look at the reward history effects for different block lengths – scale of axis in 4B ranges from -0.6 to 0.4, but is in the range -4 to 2 in 3A. Have the authors any idea why these differences arise (it could be something obvious I missed!)?

**Have all data underlying the figures and results presented in the manuscript been provided?**

Reviewer #1: Yes

Reviewer #2: **No: **There are some inadequacy of explanations of the data used.

Reviewer #3: Yes

PLOS authors have the option to publish the peer review history of their article (what does this mean?). If published, this will include your full peer review and any attached files.

Reviewer #1: No

Reviewer #2: **Yes: **Rei Akaishi

Reviewer #3: No
---

## [Decision Letter · Decision Letter 1]

10 Jun 2021

Dear  Dr Kvitsiani

Thank you very much for submitting your manuscript "Choice history effects in mice and humans improve reward harvesting efficiency" for consideration at PLOS Computational Biology. As with all papers reviewed by the journal, your manuscript was reviewed by members of the editorial board and by several independent reviewers. The reviewers appreciated the attention to an important topic. Based on the reviews, we are likely to accept this manuscript for publication, providing that you modify the manuscript according to the review recommendations.

As you will see several important issues that have been raised by both reviewer 2 and reviewer 3 remain to be fully addressed (the paper will be send back to the original reviewers).

I would also like to point out that what is claimed in this study seems to be already known in the field of experimental analysis of behaviour. Specifically, it is known that a strategy that depends on the choice history (alternating or periodic choice) maximizes the reward in the VI schedule, and that animals and humans actually adopt such a strategy when switching choices is not costly, so it would be good to further stress the novelty of the study in this respect.

Sincerely,

Stefano Palminteri

Associate Editor

PLOS Computational Biology

Daniele Marinazzo

Deputy Editor

PLOS Computational Biology

[LINK]

As you will see several important issues that have been raised by both reviewer 2 and reviewer 3 remain to be fully addressed (the paper will be send back to the original reviewers).

I would also like to point out that what is claimed in this study seems to be already known in the field of experimental analysis of behaviour. Specifically, it is known that a strategy that depends on the choice history (alternating or periodic choice) maximizes the reward in the VI schedule, and that animals and humans actually adopt such a strategy when switching choices is not costly, so it would be good to further stress the novelty of the study in this respect.

Reviewer's Responses to Questions

**Comments to the Authors:**

Reviewer #1: I am grateful to the authors for responding to my previous suggestions. However, I am still concerned about “Bayesian agent.” In addition, there is an additional issue regarding the significance of this study, which I noticed after I wrote the first review report. Since I was unable to point this out during the initial peer review, I would like to leave it to the editor to decide whether or not to take this into account.

Regarding the Bayesian agent:

In response to my previous comments regarding the Bayesian agent, the authors responded as: “The confusion in seeing the update equation as Bayesian may stem from the misspelling. The correct equation should be P(Qn(i)|R) = P(R|Qn(i)) * P(Qn(i)). We also provide additional supplementary (Fig.1 figure suppl.1C) to show that this update correctly infers the set reward probabilities.”

However, this is not enough as an answer. If one calls a Bayesian agent, the agent should represent the distribution of the variable of interest (in this case, the set reward probability). What Fig.1 suppl.1C shows are the point estimate of set reward probability rather than its distribution (By the way, the title of figure suppl.1C includes misspelling). Representing the posterior distribution requires at least making sure that its variance (uncertainty) can be expressed according to the Bayes formula. In the authors’ framework, the distribution of the estimates for the set reward probability, \\theta^n(i), should obey the posterior distribution. However, the authors did not evaluate this. When we are interested only in point estimator, we may call it Bayesian estimation if it provides the maximum a posteriori (MAP) estimator. However, it is unclear whether this is the case (whether the parameter corresponds to the MAP estimator rather than the maximum likelihood estimator) in the author's model.

Also, regarding adding the noise, the authors wrote as:

“Overall this results as we understand is similar to approaches used in hierarchical Bayesian models that use noise term in parameter estimation (Mathys C. et al, Front.in Human Neurosc. 2011).”

As far as I understand, the Mathys et al. approach assumed noise in the generation process, but the estimation is done deterministically under variational approximation, which is quite different from adding noise to the estimates as the authors did.

The Bayesian inference model is a concept that is becoming well established in behavioral modeling. Calling the author's model a Bayesian agent may cause confusion. If the author cannot properly construct a Bayesian inference model, then the Bayesian agent model should be removed from the manuscript, or the name should be changed.

Additional issue regarding the significance of the paper:

I noticed that Houston and McNamara (1981, Journal of the experimental analysis of behavior, 35(3), 367-396), which was also cited in the manuscript, theoretically showed that the periodic deterministic choice behavior gives the highest reward in the concurrent VI schedule. In addition, periodic (alternation) choice strategy have been commonly observed in VI-schedules without change over delay (COD) procedure, which seems absent in the authors’ experiment, as stated as follows in Corrado et al. (Corrado, G. S., Sugrue, L. P., Seung, H. S., & Newsome, W. T. (2005). Journal of the experimental analysis of behavior, 84(3), 581-617.):

“A second feature that our foraging task shares with many classical matching paradigms is the incorporation of a changeover delay (COD). The COD is a common technique for introducing a ‘‘cost,’’ in this case a temporal delay, to switching from one choice option to another (Shahan & Lattal, 1998). Thus, although graded matching behavior can be observed without the use of a COD (see, e.g., Lau & Glimcher, 2005), its incorporation shields the data from partial contamination with competing behavioral strategies based on alternation. Without such a cost, an animal can gather rewards surprisingly efficiently by alternating between options…”

Thus, I think that it is already known that alternating behavior, which appeared as the negative history effect of the previous trial observed in the authors' results, maximizes reward under certain VI schedule conditions. Given all this, what new knowledge can we say that the results of this paper bring? I would like to see a discussion of COD and the author's contribution in light of these previous studies.

Minor points:

Line 282:

“The higher the inverse temperature, the more stochastic the choices are…”

- I think this is the opposite.

Line 357:

were significantly different (p = 0.0006,…

- “were significantly different from zero” or “were significant”?

Line 446

“,Here an agent takes action a duringin each trial t.”

- The sentence is broken.

Reviewer #2: Authors has addressed the previous comments sufficiently. The manuscript improved with the borader perspective on this issue. Because the previous manuscript was so unfocused and confusing with the methodology, I could not see some central points of the paper. With the further analyses in this version, there are some new finding that is relevant the central claims of the paper such as the finding of the effect of the no-reward for the immediately preceding choice. So I would like to ask the authors to address the following points:

1. Can we still call the model a "double" trace model of choice history effects given that the fast component is based on no-reward on the previous choice? 

2. Choice history effect as rational or irrational bias depends on its relevance to the task. Clearly in this task the elapsed time (trials and number of choices on the options) is directly connected to the possible reward on another option. In contrast,  in other studies such as Akaishi et al. (2014), Fritsche et al. (2017). and Akrami et al.(2018), the choice history bias is irrelevant to the task performance. I am just wondering whether the interpretation of the choice history bias in the current task may not generalize to other tasks with the task irrelevance of the biases.

3. This is relatively minor point but this necessary for good scholarship with sufficient documentation of previous literature. The neural bases of the multiscale representation of the past choices have been reported. For example, single unit activities in the prefrontal cortex have been found to be related  with the fast and slow choice biases.

Mochol, Kiani, Moreno-Bote

Current Biology, 2021Prefrontal cortex represents heuristics that shape choice bias and its integration into future behavior

https://doi.org/10.1016/j.cub.2021.01.068

One of the earliest studies of choice history bias in the past decade (Akaishi et a., 2014, Neuron), from which authors burrowed the equation of the double trace model, has reported the extensive evidence about the neural bases of the multiscale choice history bias. The study described not only the involvement of the prefrontal but also the medial parietal areas and its interaction with the prefrontal cortex.

Reviewer #3: my comments have all been addressed.

**Have the authors made all data and (if applicable) computational code underlying the findings in their manuscript fully available?**

Reviewer #1: Yes

Reviewer #2: Yes

Reviewer #3: Yes

PLOS authors have the option to publish the peer review history of their article (what does this mean?). If published, this will include your full peer review and any attached files.

Reviewer #1: No

Reviewer #2: **Yes: **Rei Akaishi

Reviewer #3: No

Figure Files:

Data Requirements:

Reproducibility:

References:

---

## [Decision Letter · Decision Letter 2]

10 Aug 2021

Dear Dr Kvitsiani, 

Thank you very much for submitting your manuscript "Choice history effects in mice and humans improve reward harvesting efficiency" for consideration at PLOS Computational Biology. As with all papers reviewed by the journal, your manuscript was reviewed by members of the editorial board and by several independent reviewers. The reviewers appreciated the attention to an important topic. Based on the reviews, we are likely to accept this manuscript for publication, providing that you modify the manuscript according to the review recommendations.

As you can see Reviewer 1 is happy with the revised version of the manuscript. However, Reviewer 2 raises few remaining minor issues that should be addressed. Considering that other readers may share the same views, it could be better to clarify even more the justification of the model labelling and (also to highlight the novelty of the paper) the novelty of the model within the broader literature using the VI task or others (e.g., the references suggested by R2).

Sincerely,

Stefano Palminteri

Associate Editor

PLOS Computational Biology

Daniele Marinazzo

Deputy Editor

PLOS Computational Biology

[LINK]

As you can see Reviewer 1 is happy with the revised version of the manuscript. However, Reviewer 2 raises few remaining minor issues that should be addressed. Considering that other readers may share the same views, it could be better to clarify even more the justification of the model labelling and (also to highlight the novelty of the paper) the novelty of the model within the broader literature using the VI task or others (e.g., the references suggested by R2).

Reviewer's Responses to Questions

**Comments to the Authors:**

Reviewer #1: I think that the authors addressed my comments. I am now happy to recommend publication.

Reviewer #2: The authors responded to the questions of the reviewers partially in the new submission of this manuscript. In this resubmission, the authors added a critical piece of the new data regarding the "double trace model of choice history effect". In the previous round of the review, I have asked the authors the justifications of the use of the  "double trace model of choice history effect". Below I reproduce the interaction:

My comment: With the further analyses in this version, there are some new findings that are relevant to the central claims of the paper such as the finding of the effect of the no-reward for the immediately preceding choice. So I would like to ask the authors to address the following points:

Major point 1. Can we still call the model a "double" trace model of choice history effects given that the fast component is based on no-reward on the previous choice?

Author's response: As long as we explain what behavioral processes double trace captures, we would prefer to keep the term “double trace” as it is. 

My point was: if the alternative models incorporate both history of reward-choice interaction and history of choices, which is the implication of the new results, can we still call it the "double trace model of choice history effect"? There are confusions of the term "double trace model" throughout the paper regarding the components of the "double trace model". If it is purely consisting of choice history effects, please clearly mention this. And if the author means this, the  issue posed by the new results of "the effect of the no-reward for the immediately preceding choice" has to be resolved.

Usually the authors need to answer the questions of the reviewers not with the "preference" but "justification". It is a long review process but it is the responsibility of the authors to keep a good manner of interactions. 

The editor in the previous round of the review asked the authors to clarify the novelty of the paper given that VI task has been used in the past studies. The authors need to address this point sufficiently as well.

Also, within the discussion, in the main paragraph of page 27 the authors say "Indeed, in many decision-making tasks that do not impose contingencies of previous choices on current choices, choice history effects persist (Akaishi et al., 2014; Fritsche et al., 2017; Akrami et al., 2018; Hwang et al., 2017; Mochol, Kiani, & Moreno-Bote, 2021). Understanding the behavioral mechanisms that generate choice-history effects in these tasks may require the manipulation of behavioral contingencies, linking past choices to current choices in a parametric way." Authors can mention the papers that discuss how experimental instructions may make changes of choice history effects, or if there is any review comparing the size of choice history effects across different paradigms (foraging tasks with "replenishing" patches vs tasks with static reward probabilities, for instance). In the paradigms of perceptual decisions, (trial) frequencies of each stimulus condition in a block can bias the decisions (and possibly choice history effects too) such as the following study:

Elapsed Decision Time Affects the Weighting of Prior Probability in a Perceptual Decision Taskhttps://www.jneurosci.org/content/31/17/6339.short 

The comparison to these papers clarify the value of the current paper in the background related studies.

Some minor/technical points

It is still not clear why the authors conclude that participants assume uniform distributions of set reward probabilities. In the current version of the manuscript, the end of the first paragraph of page 12 refers the reader to two figures. But these figures are not that self-explanatory to me. I think a short explanation in that paragraph would help. Particularly because that is brought up again in the discussion, and the information there is not sufficient either.

**Have the authors made all data and (if applicable) computational code underlying the findings in their manuscript fully available?**

Reviewer #1: None

Reviewer #2: **No: **I cannot confirm now.

PLOS authors have the option to publish the peer review history of their article (what does this mean?). If published, this will include your full peer review and any attached files.

Reviewer #1: No

Reviewer #2: **Yes: **Rei Akaishi

Figure Files:

Data Requirements:

Reproducibility:

References:

---

## [Editor Report · Decision Letter 3]

15 Sep 2021

Dear Dr Kvitsiani

We are pleased to inform you that your manuscript 'Choice history effects in mice and humans improve reward harvesting efficiency' has been provisionally accepted for publication in PLOS Computational Biology.

Best regards,

Stefano Palminteri

Associate Editor

PLOS Computational Biology

Daniele Marinazzo

Deputy Editor

PLOS Computational Biology

---

## [Editor Report · Acceptance letter]

28 Sep 2021

PCOMPBIOL-D-21-00059R3 

Choice history effects in mice and humans improve reward harvesting efficiency

Dear Dr Kvitsiani,

I am pleased to inform you that your manuscript has been formally accepted for publication in PLOS Computational Biology. Your manuscript is now with our production department and you will be notified of the publication date in due course.

With kind regards,

Zsofi Zombor
